# Reversible Myc hypomorphism identifies a key Myc-dependency in early cancer evolution

Nicole M. Sodir [1,7,9] ✉, Luca Pellegrinet [1,9], Roderik M. Kortlever [1], Tania Campos[1], Yong-Won Kwon[2], Shinseog Kim [3], Daniel Garcia[4], Alessandra Perfetto[1], Panayiotis Anastasiou[1], Lamorna Brown Swigart [5], Mark J. Arends [6], Trevor D. Littlewood [1] & Gerard I. Evan [1,8] ✉

Germ-line hypomorphism of the pleiotropic transcription factor Myc in mice, either through *Myc* gene haploinsufficiency or deletion of *Myc* enhancers, delays onset of various cancers while mice remain viable and exhibit only relatively mild pathologies. Using a genetically engineered mouse model in which Myc expression may be systemically and reversibly hypomorphed at will, we asked whether this resistance to tumour progression is also emplaced when Myc hypomorphism is acutely imposed in adult mice. Indeed, adult Myc hypomorphism profoundly blocked KRas[G12D]-driven lung and pancreatic cancers, arresting their evolution at the early transition from indolent pre-tumour to invasive cancer. We show that such arrest is due to the incapacity of hypomorphic levels of Myc to drive release of signals that instruct the microenvironmental remodelling necessary to support invasive cancer. The cancer protection afforded by long-term adult imposition of Myc hypomorphism is accompanied by only mild collateral side effects, principally in haematopoiesis, but even these are circumvented if Myc hypomorphism is imposed metronomically whereas potent cancer protection is retained.

Cancers are extremely heterogeneous diseases that arise through progressive accumulation in somatic clades of diverse mutations that perturb signalling networks regulating cell growth, differentiation, repair, survival and movement. In normal cells, these same signalling networks preside over tissue ontogeny, maintenance and repair and have consequently evolved to be inherently robust. This robustness is reflected in the multiplicity, diversity and functional redundancy of the driver mutations that underpin spontaneous cancers, and in the adaptability and evolvability of cancer cells that thwart the pharmacological selective pressures we impose upon them. Nevertheless,

substantial evidence now suggests that the net output of these robust, redundant signalling networks is ultimately funnelled through a small number of non-redundant downstream effectors. One of these is Myc, a highly pleiotropic basic helix−loop−helix leucine zipper transcription factor that serves as a central, obligate and non-redundant conduit, transcriptionally coordinating the diverse intracellular and extracellular programmes required for orderly cell and tissue growth and repair[1–5].

As a pivotal arbiter of cell proliferation, Myc harbours great oncogenic potential. However, Myc activity in normal cells is tightly

---

[1]Department of Biochemistry, University of Cambridge, Cambridge CB2 1GA, UK. [2]Abcam, 860 Auburn Ct, Fremont, CA 94538, USA. [3]Center for Genomic Integrity, Institute for Basic Science, Ulsan 44919, Republic of Korea. [4]Oncogenesis Thematic Research Center at Bristol Myers Squibb, San Diego, CA 92121, USA. [5]Department of Laboratory Medicine, University of California, San Francisco, CA 94115, USA. [6]Division of Pathology, Cancer Research UK Edinburgh Centre, University of Edinburgh, Edinburgh, Scotland, UK. [7]Present address: Genentech, Department of Translational Oncology, South San Francisco, CA 94080, USA. [8]Present address: The Francis Crick Institute, NW1 1AT London, UK. [9]These authors contributed equally: Nicole M. Sodir, Luca Pellegrinet. ✉e-mail: sodirn@gene.com; gerard.evan@crick.ac.uk

restrained by negative feedback transcriptional autoregulation, which limits the peak intracellular level of Myc[6–8], and by multiple tiers of auto-attenuation that constrain Myc persistence: both Myc mRNA and protein are very short-lived, expressed only in receipt of mitogenic signalling, and are rapidly cleared from cells upon cessation of such signals[9,10]. However, in virtually all cancers this tight quantitative and temporal regulation is disrupted by various mechanisms, such as amplification of the c-myc gene (hereafter *Myc*) or its regulatory enhancers, chromosomal translocation, stabilising mutation or, most commonly, relentless induction by "upstream" oncogenic drivers[11]. These all result in aberrant elevation and persistence of Myc, both of which are thought to contribute to its oncogenic activity. Many studies in diverse cancer models concur that Myc activity is essential for maintenance of most cancers, irrespective of their cell of origin or underlying oncogenic mechanism and, importantly, whether or not Myc itself is a direct oncogenic "driver."

In addition to its generic role in tumour maintenance, evidence also suggests that the level of endogenous Myc expression is a critical determinant of cancer incidence. In normal cells, Myc levels are governed by a plethora of evolutionarily conserved upstream and downstream enhancer and super-enhancer elements that tailor Myc expression to the specialised developmental, maintenance and regenerative needs of each tissue and organ. Nonetheless, certain naturally occurring allelic variations within these regulatory regions can generate cryptic recognition elements for lineage-specific transcription factors, augmenting Myc expression and greatly predisposing to various cancers[12–18]. Topological disruption[19,20] or amplification of *Myc* enhancers[21,22] is also observed in certain cancer types. Conversely, germline ablation of many of these evolutionarily conserved cancer-predisposing regulatory elements markedly decreases incidence of associated cancers while eliciting remarkably little impact on normal mouse development or adult tissue physiology[13,21,23,24]. In many instances such cancer protection correlates with reduced basal levels of Myc expression in the affected organ[13] suggesting that reduced basal Myc expression is cancer-protective. Myc haploinsufficiency, which also reduces basal Myc expression in many tissues[25,26], has likewise been shown to confer substantial protection against onset of intestinal polyposis in *APC^Min* mice[27–29], spontaneous lymphoma[30] and KRas^G12D-induced pancreatic adenocarcinoma[31]. Moreover, immortalised *Myc* haploinsufficient fibroblasts show marked resistance to Ras or Raf transformation in vitro[32] as well as reduced global protein and ribosome synthesis, extension of G1 and G2 cell cycle phases[33], and an increased tendency to upregulate the CDK4/6 inhibitor p16^INK4 and engage replicative senescence[34].

The remarkable delay in onset of certain cancers afforded by global reduction of endogenous Myc suggests the existence of a bottleneck, sometime in early tumour evolution, whose transit is dependent on exceeding a critical threshold level of Myc expression, above that needed for most quotidian physiological Myc functions. One possible explanation for this is that Myc operates in two physiological regenerative modes. In habitually regenerating tissues, such as GI tract and bone marrow, epithelial cell proliferation takes place within pre-established somatic niches and the principal role of Myc is to coordinate cell-autonomous transcriptional programmes that are needed for cell division. By contrast, regenerative repair following trauma or disruptive infection requires Myc to coordinate the far more extensive role of niche reconstruction[35], reprogramming local ECM from its dense, quiescent, basement membrane-configured state into a dynamically niche-rebuilding mode through coordinated choreography of inflammatory, immune, mesenchymal and endothelial stromal cells. Similarly, Myc plays a pivotal role in driving the transition from pre-tumour to adenocarcinoma by directing release of tissue-specific cocktails of signals that instruct ECM reprogramming[36–38]. It seems plausible that both Myc's physiological reparative mode and early tumour progression require higher levels of Myc activity and are consequently especially sensitive to Myc hypomorphism.

In mice, the tumour-protective benefit of germline Myc haploinsufficiency is accompanied by unwelcome side-effects, including small body size with variably reduced cellularity across organs[26], infertility[25] and a profound deficit in haematopoiesis due to aberrant self-renewal of haematopoietic stem cells[39]. However, the extent to which these diverse pathologies and reduced cancer incidence are direct consequences of Myc insufficiency versus indirect developmental compensation for Myc insufficiency in the embryo or placenta is unknown. Yet, exploiting long-term Myc suppression in cancer prevention is feasible only if such prophylaxis is dissociable from accompanying pathologies and achievable by systemic Myc inhibition in adult life. Here, we address these issues using a unique mouse model in which endogenous Myc hypomorphism may be reversibly induced and relieved at will.

## Results

In mice, germline hypomorphism of Myc delays the onset of many tumour types. However, the mechanism underlying such cancer prevention remains unclear, as does whether it is a direct result of Myc reduction or an indirect consequence of adaptive compensation during development. To determine whether acute imposition of Myc hypomorphism in adult animals affords analogous cancer protection, and at what stage of cancer evolution such protection acts, we generated a mouse in which endogenous Myc expression may be systemically and reversibly hypomorphed at will. We inserted a heptameric tetracycline-response element (*TRE*) into the second intron of the endogenous c-myc gene (Supplementary Fig. 1a). The resulting *Myc^TRE/TRE* (*M*) mice were then crossed with our *tTS^Kid* repressor strain (*R*) in which the tetracycline-dependent tTS^Kid repressor is ubiquitously and constitutively expressed from the β-actin promoter[40], so generating *Myc^TRE/TRE*; *tTS^Kid/–* (hereafter called *MR*) mice. In the absence of tetracycline, the tTS^Kid repressor binds to the 2nd intron *TRE*, partially repressing *Myc* expression. Administration of tetracycline rapidly and reversibly relieves this partial repression, so generating a reversibly switchable Myc hypomorph (Supplementary Fig. 1b). Insertion of the *TRE* element into the endogenous c-myc 2nd intron had no detectable impact on either levels or the signature transient kinetics of Myc induction by serum[41] in embryonic *MR* fibroblasts (Supplementary Fig. 2).

To validate the efficacy and reversibility of Myc hypomorphism in *MR* mice in vivo, we determined levels of endogenous Myc expression in a variety of adult tissues that normally proliferate (and consequently demonstrate measurable basal Myc expression) - bone marrow, spleen, thymus and small intestine. Wild type levels of Myc were maintained in *MR* mice throughout gestation and into adult life by sustained administration of tetracycline. Tetracycline was then withdrawn to activate the *tTS^Kid* repressor and *Myc* RNA levels assessed by RT-PCR over the ensuing 4 weeks. Representative data for small intestine, spleen, thymus and bone marrow (Supplementary Fig. 3a) confirmed a profound, stable and reproducible decrease in endogenous *Myc* expression of between 25 and 50%, depending on tissue, that was completely reversed within 1 week upon re-addition of tetracycline. We guess the modest variability in Myc reduction across different tissues reflects cell type variations in intrinsic activity of the β-actin "house-keeping" promoter we used to drive transgenic expression of tTS^Kid[42]. During the 4-week test period of induced systemic Myc hypomorphism we observed no detectable deleterious impact on animal health, activity or welfare, or tissue architecture (Supplementary Fig. 3b), nor any significant perturbation in blood cell counts (Supplementary Fig. 3c).

Determining the extent of inducible Myc hypomorphism in most adult cancer-prone *MR* mouse tissues, such as lung and pancreas, is complicated by the fact that they are, in the main, non-proliferative

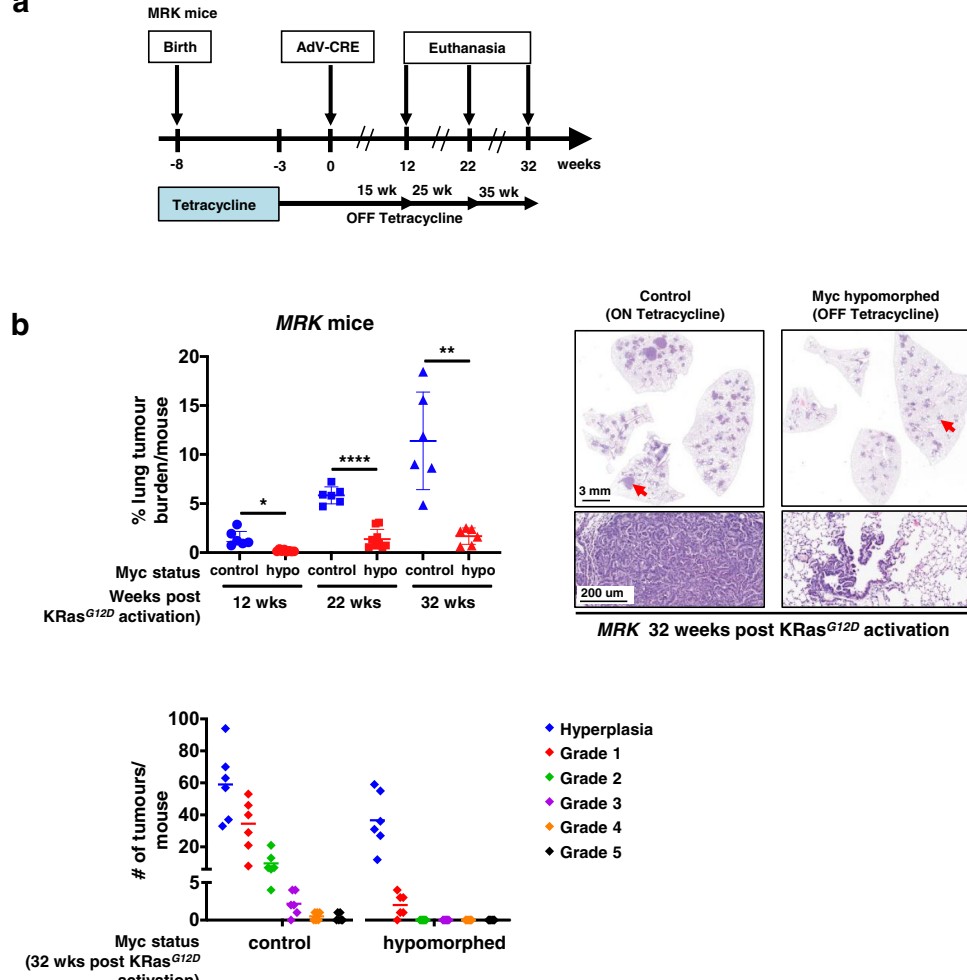

**Fig. 1 | Imposition of Myc hypomorphism in adult mice blocks progression of KRas^G12D-driven lung hyperplasia to adenocarcinoma. a** Schematic of study. Endogenous Myc was maintained at normal levels in *MRK* (*Myc^TRE/TRE*;β-actin-tTS^Kid/−^;*LSL-KRas^G12D/+*) mice through development, neonatal growth and into adulthood (5 weeks old) by continuous administration of tetracycline. Endogenous Myc was then hypomorphed by withdrawal of tetracycline and three weeks later Adv-Cre administered by inhalation to trigger sporadic expression of KRas^G12D in lung epithelium. Mice were then euthanized 12, 22, or 32 weeks later. Control mice were treated identically save that tetracycline was maintained throughout. **b** Top left: percentage of tumour burden relative to total lung in non-hypomorphed (blue) versus hypomorphed (red) *MRK* mice at 12, 22, or 32 weeks post Adv-Cre activation of *KRas^G12D*. Results depict mean ± SD in each treatment group. The unpaired t-test with Welch's correction was used to analyse tumour burden. *p < 0.05, **p < 0.01, ****p < 0.0001. SD = standard

deviation. For 12 weeks: *n* = 6 for non-hypomorphed control and *n* = 8 for hypomorphed mice with *p* = 0.0115; for 22 weeks: *n* = 6 for non-hypomorphed control and *n* = 9 for hypomorphed mice with *p* < 0.0001; for 32 weeks: *n* = 6 for both non-hypomorphed control and hypomorphed mice with *p* = 0.0045. Top right: Representative H&E-stained sections of lungs from *MRK* mice at 32 weeks post Adv-Cre activation of *KRas^G12D* either non-hypomorphed (maintained on tetracycline) or hypomorphed (off tetracycline). Arrows mark regions shown at higher magnification below. Bottom left: Grading of tumours (after[43]) in lungs from *MRK* mice at 32 weeks post Adv-Cre activation of *KRas^G12D*, either non-hypomorphed (control, maintained on tetracycline) or hypomorphed (off tetracycline). Results depict quantitation of total numbers of tumours of each grade, with the means indicated. Source data are provided as a Source Data file.

and consequently express negligible basal levels of endogenous Myc. However, both lung and pancreas proliferate and grow extensively during early postnatal life and consequently express endogenous Myc. We therefore assessed the extent of inducible Myc hypomorphism in neonatal *MR* lung and pancreas. *MR* embryos were allowed to develop with normal Myc levels and tetracycline then withdrawn at birth to hypomorph endogenous Myc. 8 and 14 days later, neonates were euthanized, lungs and pancreata harvested, and endogenous *Myc* mRNA levels assessed (Supplementary Fig. 4a). Relative to controls (animals maintained on tetracycline), lung and pancreas of hypomorphed neonatal mice reproducibly exhibited Myc mRNA repression (between 30 and 60%) at both neonatal time points (Supplementary Fig. 4b).

Since germline Myc haploinsufficiency significantly delays onset of tumours in various cancer mouse models, we next asked whether

acute imposition of Myc hypomorphism in adult mice is similarly protective. First, we crossed *MR* mice into the well-characterised *LSL-Kras^G12D* mouse model of lung adenocarcinoma (LUAD)[43] to generate adult *MRK* mice. Myc was then hypomorphed (tetracycline-deprived) for 3 weeks in adult mice prior to sporadic activation of KRas^G12D in lung epithelium by Adenovirus-Cre inhalation. Thereafter, the mice were maintained in their hypomorphic state and tumour burden and progression relative to non-hypomorphed controls assessed 12, 22 or 32 weeks later (Fig. 1a). Prior studies have confirmed that the multiple discrete pre-neoplastic lesions generated by Adenovirus-Cre inhalation are each an independent oncogenic event[44] so that each mouse in effect harbours multiple independent lung pre-tumours.

As expected[43], by 12 weeks post KRas^G12D activation non-hypomorphed control mice developed multiple hyperplastic (AAH) lung foci. Thereafter, tumour burden progressively grew along with

rapidly increasing representation of high grade, aggressive disease, including adenocarcinomas. By contrast, while Myc hypomorphed mice presented similar initial numbers of AAH lung lesions as their early non-hypomorphed controls, subsequent tumour progression was almost completely blocked (Fig. 1b).

Inactivation of p53 greatly accelerates KRas$^{G12D}$-driven LUAD progression[45]. We therefore next asked whether p53 inactivation negates the prophylactic impact of Myc hypomorphism on lung tumourigenesis. *MRK* mice were crossed into a *p53^{flox/flox}* background to generate *MRKP^{fl}* mice, so allowing for concurrent deletion of p53 in lung epithelia along with activation of KRas$^{G12D}$ upon Adenovirus-Cre inhalation. The previous hypomorph tumour prophylaxis experiment was then repeated. As reported[45], p53 inactivation greatly accelerated lung tumour progression in the non-hypomorphed control *MRKP^{fl}* animals relative to their p53 *wt* counterparts, requiring them to be culled after only 14 weeks. Nonetheless, Myc hypomorphism once again potently retarded global tumour formation and markedly reduced tumour load (Fig. 2a). However, in addition to the abundant background of stalled AAH lesions, similar to that observed in hypomorphed p53^{wt} *MRK* animals, we noted the sporadic emergence of a few large, invasive lung tumours. Investigation revealed that almost all of these "escapee" tumours had circumvented our Myc hypomorphism switch, either through loss of the tTS$^{Kid}$ repressor expression or by gross up-regulation of endogenous Myc, or both (Supplementary Fig. 5a). Hence, p53 inactivation does not of itself negate the cancer protection afforded by hypomorphing Myc although it does increase the probability of happenstance genetic mayhem breaking our experimental mouse model.

Our lung cancer models indicate that imposition of Myc hypomorphism prior to KRas$^{G12D}$ activation is highly protective against the evolution of invasive lung cancer, forestalling the transition from indolent hyperplasia to invasive disease. To address whether Myc hypomorphism imposed *after* KRas$^{G12D}$ activation also suppresses progression of precancerous lesions, we used the well-characterised *LSL-KRas^{G12D/+};LSL-p53^{R172H/+};pdx1-Cre;* (*KPC*) mouse model of pancreatic adenocarcinoma (PDAC)[46]. In this model, KRas$^{G12D}$ is activated (alongside the p53$^{R172H}$ mutant) early in embryogenesis (around E8) in pancreatic and duodenal progenitor cells by Cre recombinase driven from the *pdx/IPF1* (*Pancreas/duodenum homeobox protein 1*) promoter[46]. Adult *KPC* mice rapidly develop multiple invasive and metastatic PDACs that closely recapitulate all characteristics of their human counterparts. To ascertain the protective impact of Myc hypomorphism imposed post KRas$^{G12D}$, we crossed *KPC* mice into our *MR* background to generate *MRKPC* mice and then at 4 weeks of age withdrew tetracycline to hypomorph endogenous Myc. Tumours were allowed to develop and after 18 weeks mice were euthanized and pancreata isolated and assessed for tumour load. Pancreata of most (~85%) control (non-hypomorphed) mice exhibited an extremely high tumour load, comprising multiple invasive pancreatic adenocarcinomas. By contrast, the pancreata of hypomorphed Myc-mice showed dramatically reduced tumour burden confined almost exclusively to indolent PanINs (Fig. 2b). However, just as with the lungs of *MRKP^{fl}* mice (above), we noted occasional, isolated large and aggressive "escapee" pancreatic tumours in ~30% of Myc-hypomorphed animals. Further analysis revealed that almost all of these escapee pancreatic tumours had markedly downregulated expression of the tTS$^{Kid}$ repressor and/or upregulated expression of endogenous Myc (Supplementary Fig. 5b), thereby circumventing the hypomorphing mechanism used in our model. Thus, as in the *MRKP^{fl}* lung model previously described, p53 inactivation does not of itself negate the potent cancer prophylaxis afforded by Myc hypomorphism but it does increase the likelihood of our model's sporadic breakage. In this regard, it is noteworthy that we never observed "escapee" breakthrough lung tumours arising in p53*wt* *MRK* mice over their lifespan. This indicates that sporadic inactivation of p53, which would presumably potentiate such escape mechanisms,

does not significantly erode the protection afforded by Myc hypomorphism in p53 competent animals.

Taken together, our data indicate that adult imposed Myc hypomorphism acts as a bottleneck very early in tumour evolution, post KRas$^{G12D}$ activation but prior to the transition to overt, locally invasive cancer. To map more precisely the location of the Myc hypomorphism bottleneck in progression of lung and pancreas pre-cancers, we constructed an allelic series of knock-in mice in which different levels of the reversibly switchable, 4-hydroxytamoxifen-dependent Myc variant MycER$^{T2}$ may be ectopically activated at will in lung or pancreas epithelium[47]. We previously showed that MycER$^{T2}$ driven from two alleles of the endogenous *Rosa26* promoter (*R26^{MT2/MT2}* homozygotes) is expressed at a level broadly equivalent to that of endogenous Myc in proliferating cells and tissues[38,47,48]. By contrast, mice in which MycER$^{T2}$ expression is driven from only a single *Rosa26* allele (*R26^{MT2/+}* hemizygous) express only half as much MycER$^{T2}$[47,48], a sub-physiological level comparable to that of hypomorphed endogenous Myc in proliferating tissues of *MR* mice. As previously shown, acute activation of physiological levels of MycER$^{T2}$ from two alleles of the *Rosa26* promoter in (*LSL-KRas^{G12D/+};R26^{MT2/MT2}* lung (referred to as *KR26^{MT2/MT2}*) and *LSL-KRas^{G12D/+};pdx1-Cre; R26^{MT2/MT2}* pancreas (referred to as *KCR26^{MT2/MT2}*) is sufficient in vivo to drive oncogenic cooperation with oncogenic KRas$^{G12D}$, triggering immediate and synchronous transition of indolent KRas$^{G12D}$-driven lung pre-tumours (Fig. 3a) and PanINs (Fig. 3b) to adenocarcinomas along with all the invasive, proliferative, stromal, inflammatory, immune and vascular signature attributes of spontaneous adenocarcinomas from those same organs[37,38]. In sharp contrast, activation of the reduced levels of MycER$^{T2}$ from only a single *Rosa26* promoter (equivalent to levels of endogenous Myc in hypomorphed *MR* animals) in the indolent tumours of hemizygous *KR26^{MT2/+}* (lung) and *KCR26^{MT2/+}* (pancreas) mice elicited no discernible change in lung adenoma/PanIN morphology, proliferation or stroma (Fig. 3a, b). Previously, we demonstrated in our switchable *KR26^{MT2/MT2}* lung LUAD mouse model that activation of Myc at physiological levels drives expression and release from pre-tumour lung epithelium of two key pro-tumourigenic instructive paracrine signals whose actions are absolutely required for the transition from pre-tumour to invasive neoplasia[37]. One of these, CCL9, orchestrates the rapid influx of VEGF-expressing macrophages that triggers the critical angiogenic switch required for macroscopic tumour growth[37]. The other, IL-23, drives expulsion of NK, T and B lymphoid cells from the inchoate pre-tumour and is absolutely required for tumour outgrowth and viability[37]. Histological analysis confirmed that activation of physiological levels of Myc in *KR26^{MT2/MT2}* lung pre-tumours triggered the expected rapid CCL9-dependent influx of CD206$^+$ macrophages, induction of IL-23 and reduction in CD3$^+$ T cells and NKp46$^+$ NK cells. By contrast, activation of reduced, hypomorphic-equivalent, levels of deregulated Myc from a single *Rosa26* promoter did neither and elicited no discernible reprogramming of the local somatic microenvironment (Fig. 4). Nonetheless, such reduced levels of ectopic Myc retained measurable transcriptional activity and still induced a significant, but selective, fraction of the Myc target genes induced in their non-hypomorphed counterparts, albeit to a generally lesser and more selective degree (Supplementary Fig. 6). These data indicate that hypomorphic levels of Myc are insufficient to support the transition from pre-tumour to invasive tumour because they fail to drive release of key, instructive stromal modifying signals at a level (or a duration) required to programme a microenvironment conducive to macroscopic tumour outgrowth and invasion.

This inability of hypomorphic Myc to support tumour development raises the possibility that long-term reduction in Myc activity in adults, perhaps imposed by some future Myc-inhibitor drug, might be powerfully prophylactic against cancers. The obvious concern, however, is that such reduced Myc might prove insufficient to maintain key homeostatic roles that Myc serves, perhaps during *in utero*

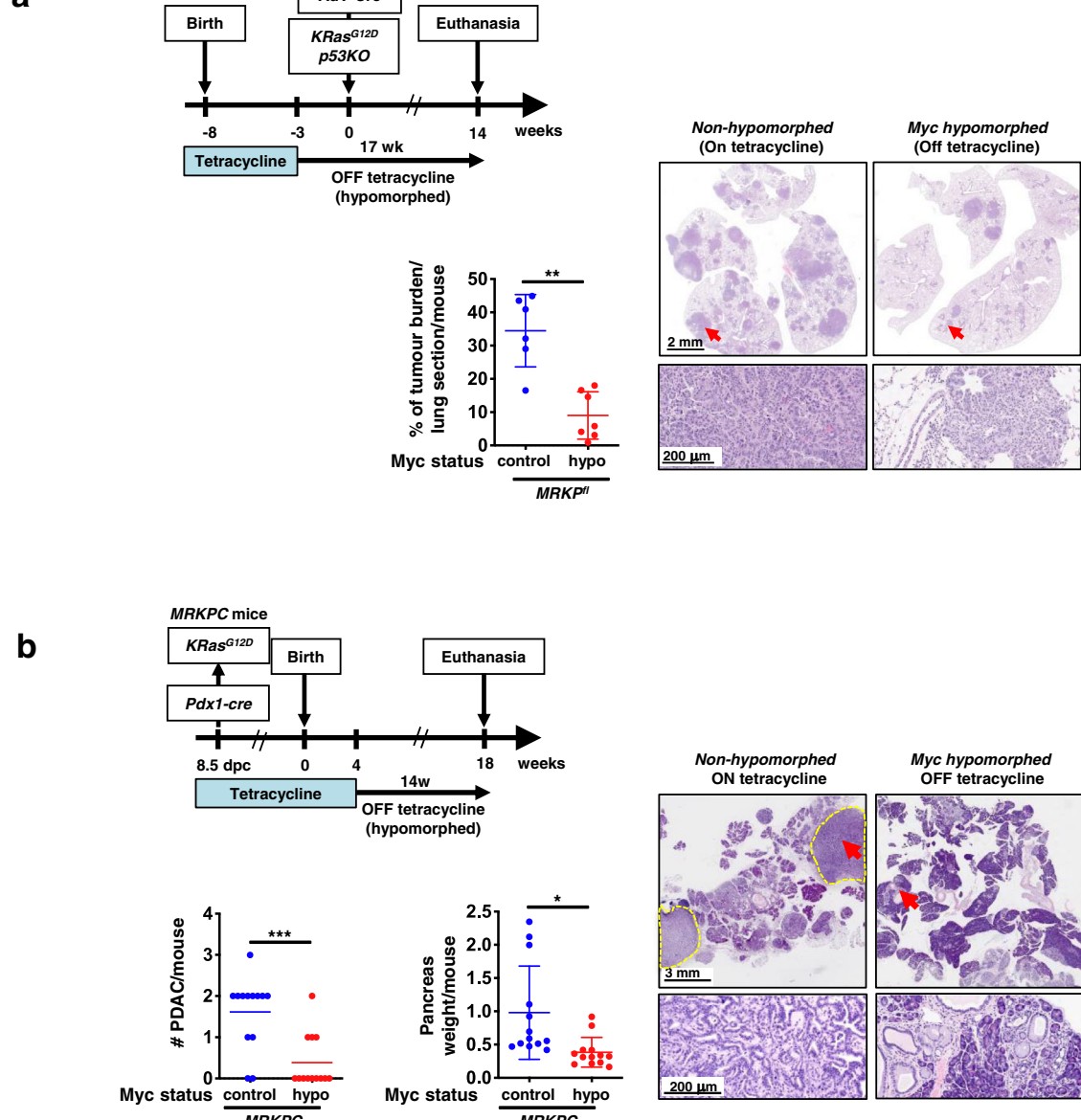

**Fig. 2 | Myc hypomorphism blocks lung and pancreatic tumour progression post KRas^{G12D} activation but before the transition from indolent pre-tumour to invasive neoplasia. a** Left: schematic of study. *MRKP^{fl}* (*Myc^{TRE/TRE}*;*β-actin-tTS^{Kid/−}*;*LSL-KRas^{G12D/+}*;*p53^{flox/flox}*) mice were maintained on tetracycline (endogenous Myc at *wt* levels) throughout embryonic and post-natal development until 5 weeks of age. Tetracycline was then withdrawn to hypomorph Myc and 3 weeks later KRas^{G12D} activated and p53 concurrently inactivated sporadically in lung epithelium by Adv-Cre inhalation. Mice were euthanized 14 weeks post Adv-Cre inhalation and lung tissues harvested. Control animals were treated identically save that they were maintained throughout on tetracycline to sustain *wt* endogenous Myc levels. Centre: quantitation of overall tumour load in non-Myc hypomorphed (blue) versus hypomorphed (red) *MRKP^{fl}* mice 14 weeks post Adv-Cre inhalation. Results depict mean ± SD in each treatment group. The unpaired *t*-test with Welch's correction and two-tailed analysis was used to analyse tumour load. **$p < 0.01$. $n = 6$ for non-Myc hypomorphed and $n = 7$ for hypomorphed mice with $p = 0.0010$. Right: representative H&E staining of lung tissue harvested from non-hypomorphed versus hypomorphed *MRKP^{fl}* mice 14 weeks post Adv-Cre inhalation. Arrows mark regions shown at higher magnification below. **b** Upper left: Schematic of study. In *MRKPC* (*Myc^{TRE/TRE}*;*β-actin-tTS^{Kid/−}*;*LSL-KRas^{G12D/+}*;*LSL-p53^{R172H/+}*;*Pdx-1-Cre*) mice

expression of Cre recombinase is driven from the *pdx/IPF1* promoter, triggering co-expression of KRas^{G12D} and p53^{R172H} in pancreatic progenitor cells from around 8.5 *dpc*. *wt* levels of endogenous Myc were maintained throughout development and into adulthood by continuous administration of tetracycline. At 4 weeks of age, tetracycline was withdrawn from mice to hypomorph endogenous Myc and animals euthanized 14 weeks later (total 18 weeks old). Tetracycline administration was maintained throughout in the control, non-hypomorphed cohort. Bottom left: quantitation of PDAC tumours per mouse in non-hypomorphed (blue) versus hypomorphed (red) *MRKPC* mice and overall pancreas weight (a surrogate for tumour load) in the same animals. The unpaired t-test with Welch's correction and two-tailed analysis was used to analyse the data. Mean ± SD are shown. *$p < 0.05$, ***$p < 0.001$. FOV = field of view. SD = standard deviation. For quantitation of PDAC tumours and overall pancreas weight, $n = 13$ for both non-Myc hypomorphed control and hypomorphed mice with $p = 0.0005$ and $p = 0.0110$, respectively. Right: representative macroscopic, low- and high-power images of non-hypomorphed versus hypomorphed pancreata at 18 weeks, showing multiple PDAC tumours in the former but none in the latter. Dotted yellow lines represent the margins of a representative PDAC tumour. Arrows mark regions shown at higher magnification below. Source data are provided as a Source Data file.

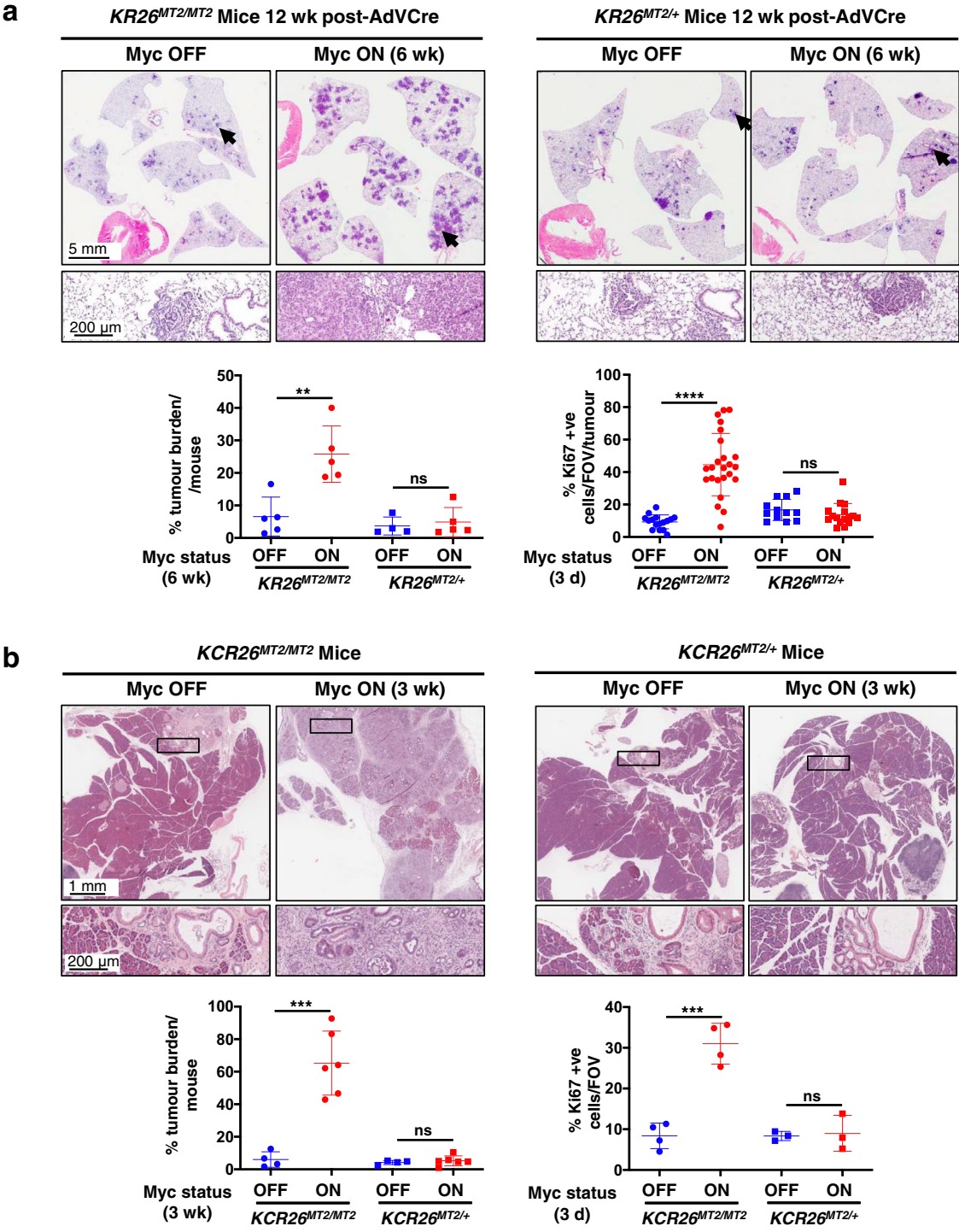

development, neonatal growth, and maintenance and repair of adult tissues. We therefore sought pathologies associated with acutely imposed hypomorphism in *MR* mice. Since germline *Myc*-deficient embryos fail around 10.5 *dpc*[25,26,49,50] we first asked whether imposed hypomorphism of endogenous Myc is compatible with embryonic development. *Myc^{TRE/TRE}* (*M*) females were mated with *MR* (*Myc^{TRE/TRE}*;*tTS^{Kid/−}*) males and maintained either with tetracycline (i.e. Myc at *wt* levels) or without (i.e. Myc hypomorphed). *MR* embryos carried by *M* mothers and maintained at *wt* Myc levels were born at a normal Mendelian ratio and developed normally, as did their *M* (*Myc^{TRE/TRE}* but *tTS^{Kid}*-negative) control littermates. By contrast, *MR* embryos maintained from conception as Myc hypomorphs all failed and were resorbed by 13.5 *dpc* (Supplementary Fig. 7a, b). As with Myc-deficient

embryos, developmental failure was associated with a range of developmental deficits, most obviously in vasculogenesis. Unexpectedly, such mid-gestation lethality was only evident when the breeding male, but not the female, contributed the *tTS^{Kid}* repressor (i.e. ♀*M* x ♂*MR*). In crosses where the *tTS^{Kid}* repressor allele was carried by the mother (i.e. ♀*MR* × ♂*M*), tetracycline removal triggers Myc hypomorphism in both embryos and their mothers and this resulted in immediate failure and resorption of *all* embryos – presumably both those that did not carry *tTS^{Kid}* repressor gene along with those that did. Such catastrophic early developmental failure must a consequence of Myc hypomorphism in the mothers, not their embryos, and most likely reflects a requirement for maximal Myc expression in early pregnancy and implantation, presumably to underpin decidualization or

**Fig. 3 | Deregulated Myc must be expressed above a tight threshold level to drive transition from indolent pre-tumour to lung and pancreas adenocarcinoma in vivo. a** Co-expression of KRas$^{G12D}$ and MycER$^{T2}$ was sporadically triggered in lung epithelium of either $KR26^{MT2/MT2}$ ($LSL$-$KRas^{G12D/+}$;$R26^{MT2/MT2}$ – 2 copies of $Rosa26$-$MycER^{T2}$) or $KR26^{MT2/+}$ ($LSL$-$KRas^{G12D/+}$;$R26^{MT2/+}$ 1 copy of $Rosa26$-$MycER^{T2}$) adult mice by Adenovirus-Cre inhalation. 6 weeks later (i.e. after 6 weeks of KRas$^{G12D}$-only activity) MycER$^{T2}$ was also activated for a further 6 weeks (Myc ON). MycER$^{T2}$ was not activated in control mice (Myc OFF). Representative H&E-stained sections of lungs from $KR26^{MT2/MT2}$ versus $KR26^{MT2/+}$ mice are shown without (left) and with (right) Myc activation. H&E panel inserts below show arrowed regions at higher magnification. Lower panel left: quantitation of percentage tumour burden relative to the total lung in $KR26^{MT2/MT2}$ and $KR26^{MT2/+}$ mice, either without or with Myc for 6 weeks. Results depict mean ± SD. Data were analysed using unpaired t-test with Welch's correction and two-tailed analysis. For $KR26^{MT2/MT2}$ mice, $n = 5$ for both Myc OFF and Myc ON groups with $p = 0.0045$; for $KR26^{MT2/+}$ mice, $n = 4$ for Myc OFF and $n = 5$ for Myc ON groups with $p = 0.6346$. Lower panel right: quantitation of proliferation (immunohistochemical staining for Ki67) of independent tumours in sections of lungs harvested from $KR26^{MT2/MT2}$ and $KR26^{MT2/+}$ mice 12 weeks after activation of KRas$^{G12D}$ either without or with activation of MycER$^{T2}$ for 3 days. Results depict mean ± SD. For $KR26^{MT2/MT2}$ mice, $n = 16$ independent tumours (4 mice) for Myc OFF and $n = 24$ independent tumours (6 mice) for Myc ON groups with $p < 0.0001$; for $KR26^{MT2/+}$ mice, $n = 12$ independent tumours (3 mice) for Myc OFF and $n = 16$ independent tumours (4 mice) for Myc ON groups with $p = 0.2553$. **b**. Representative H&E-stained sections of pancreata from 15 week-old $KCR26^{MT2/MT2}$ ($LSL$-$KRas^{G12D/+}$;$pdx1$-$Cre$;$R26^{MT2/MT2}$) and $KCR26^{MT2/+}$ ($LSL$-$KRas^{G12D/+}$;$pdx1$-$Cre$;$R26^{MT2/+}$) mice either without or with Myc activation for 3 weeks. H&E panel inserts below show boxed regions at higher magnification. Lower panel left: quantitation of percentage tumour burden relative to the total pancreas from $KCR26^{MT2/MT2}$ and $KCR26^{MT2/+}$ mice, either without or with Myc activation for 3 weeks. Results depict mean ± SD. Data were analysed using unpaired t-test with Welch's correction and two-tailed analysis. For $KR26^{MT2/MT2}$ mice, $n = 4$ for Myc OFF and $n = 6$ for Myc ON groups with $p = 0.0004$; for $KR26^{MT2/+}$ mice, $n = 4$ for Myc OFF and $n = 6$ for Myc ON groups with $p = 0.4809$. Lower panel right: quantitation of proliferation (Ki67) in sections of pancreata harvested from 12 week-old $KCR26^{MT2/MT2}$ and $KCR26^{MT2/+}$ mice, either without or with activation of MycER$^{T2}$ for 3 days. Results depict mean ± SD; each data point represents the average quantification of % ki67 staining of lesions from four different regions of the pancreas. Data were analysed using unpaired t-test with Welch's correction and two-tailed analysis. For $KR26^{MT2/MT2}$ mice, $n = 4$ for both Myc OFF and Myc ON groups with $p = 0.0006$; for $KR26^{MT2/+}$ mice, $n = 3$ for both Myc OFF and Myc ON groups with $p = 0.8248$. **$p < 0.01$, ***$p < 0.001$, ****$p < 0.0001$, ns= non-significant. SD = standard deviation. Part of Fig. 3 (some data relating to $KR26^{MT2/MT2}$ and $KCR26^{MT2/MT2}$ mice) is included for reference and is from previous publications[37,38]. Source data are provided as a Source Data file.

endometrial-blastocyst signalling in hypomorphed mothers[51]. The contraceptive phenotype was rapidly reversed upon re-administration of tetracycline to breeding *MR* females and *M* males (Supplementary Fig. 7c).

We next investigated the impact on postnatal growth and organ maturation of imposing Myc hypomorphism immediately after birth. *MR* embryos were allowed to develop with normal Myc levels and tetracycline then withdrawn at birth to hypomorph Myc. 8 and 14 days later, neonates were euthanized, lungs and pancreata harvested, and tissues examined histologically and for proliferation (Ki67) relative to non-hypomorphed controls, as previously described (Supplementary Fig. 4b). Despite substantial reductions in endogenous Myc expression at both 8 and 14 days post-partum, both organs showed completely normal architectures and proliferation indices (Supplementary Fig. 8a). Furthermore, post-partum rates of increase in total animal weight were identical in both hypomorphed and control cohorts, with both hypomorphed and control animals achieving the same weight at the same time (Supplementary Fig. 8b).

To delineate temporally the embryonic bottleneck at which Myc hypomorphism impacts mouse development, *MR* embryos were hypomorphed from mid-gestation by withdrawal of tetracycline from pregnant mothers at 13.5 *dpc* and hypomorphism maintained thereafter. Remarkably, post 13.5 *dpc* hypomorphism elicited no measurable deleterious impact on subsequent embryo development, neonatal growth or proliferation, nor any adverse impact on achieving final adult organ size (Supplementary Fig. 9). Hence, the 13.5 *dpc* mid-gestation Myc hypomorphism-sensitive bottleneck is transient.

Although no deleterious impacts were observed on homeostasis and health of adult *MR* mice in our initial 4-week study of continuous Myc hypomorphism (Supplementary Fig. 3), we nonetheless searched for pathologies arising from longer-term sustained adult Myc hypomorphism. Indeed, after around 6 weeks, we reproducibly observed onset of progressive leukopenia, followed at 12 weeks by onset of progressive erythropenia and, at around 18 weeks by onset of thrombocythemia (Supplementary Fig. 10a). We also noted modest splenic extramedullary haematopoiesis in ~50% of animals after 15 weeks (Supplementary Fig. 10b). While all affected mice remained active and appeared healthy throughout, this suggested a variably penetrant requirement for maximal expression of endogenous Myc in long-term maintenance of adult haematopoiesis. Remarkably, however, relaxation of Myc hypomorphism rapidly restored normal blood counts in all animals (Supplementary Fig. 3A). We therefore asked whether imposing Myc hypomorphism metronomically might circumvent the adverse haematopoietic disruption of chronic Myc hypomorphism while yet preserving effective cancer protection. To do this, KRas$^{G12D}$ was activated in *MRK* mouse lung epithelium by inhalation of Adenovirus-Cre recombinase and animals thereafter maintained on an alternating regimen of 4-weeks induced Myc hypomorphism followed by a 1 week reversal, for a total of 22 weeks (Fig. 5a). Metronomically hypomorphed mouse blood counts remained normal throughout (Fig. 5b) with no evidence of extramedullary splenic haematopoiesis. Nonetheless, all pre-neoplastic lesions remained stalled with negligible tumour load, in stark contrast to the fulminant tumours in lungs of control, non-hypomorphed, *MRK* mice (Fig. 5c).

## Discussion

In addition to its well-documented role in the maintenance of diverse cancer types, circumstantial evidence has mounted that Myc levels play a pivotal gatekeeper role in early evolution of cancers, irrespective of their oncogenic mechanism or cell of origin. Relatively subtle germline allelic variations and mutations in *Myc* gene enhancer regulatory elements exert a huge influence on lifetime risk of diverse cancers in mice. Most notably, germline modifications that reduce basal levels of Myc expression, such as enhancer deletion or *Myc*$^{+/-}$ haploinsufficiency, afford significant protection from a wide variety of adult-onset cancers while having surprisingly little evident impact on normal mouse development and adult tissue homeostasis[13,21,23,24,27,28,31]. Such observations are provocative in that they suggest that partial systemic inhibition of Myc, either directly by some future Myc inhibitor or perhaps by damping Myc expression indirectly, offers a credible strategy for cancer prevention. However, the mechanism by which reduced germline Myc levels impede oncogenesis, and at what stage of cancer evolution such restraint operates, remain unclear. Furthermore, many phenotypes associated with germline perturbation of specific genes are, at least in part, epistatic consequences of organism-level adaptive compensation[52–54]. Others manifest only at bottlenecks in embryonic or neonatal development but are not replicated upon equivalent genetic or pharmacological perturbation in adults. In the case of Myc, haploinsufficiency is also associated with a range of unwelcome, capriciously penetrant side-effects, including small body size with variably reduced cellularity across organs[26], infertility[25], and defective haematopoiesis due to aberrant self-renewal of haematopoietic stem cells[39].

To test directly whether de novo imposition of Myc hypomorphism in adults might have cancer-protective effects, and what accompanying deficits or iatrogenic pathologies such induced adult

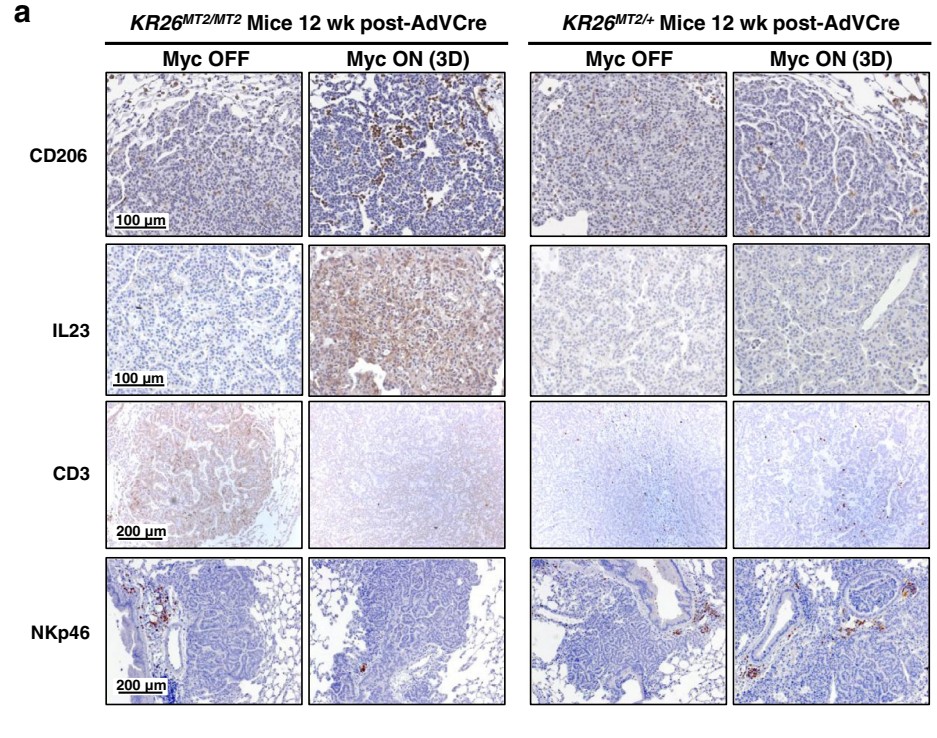

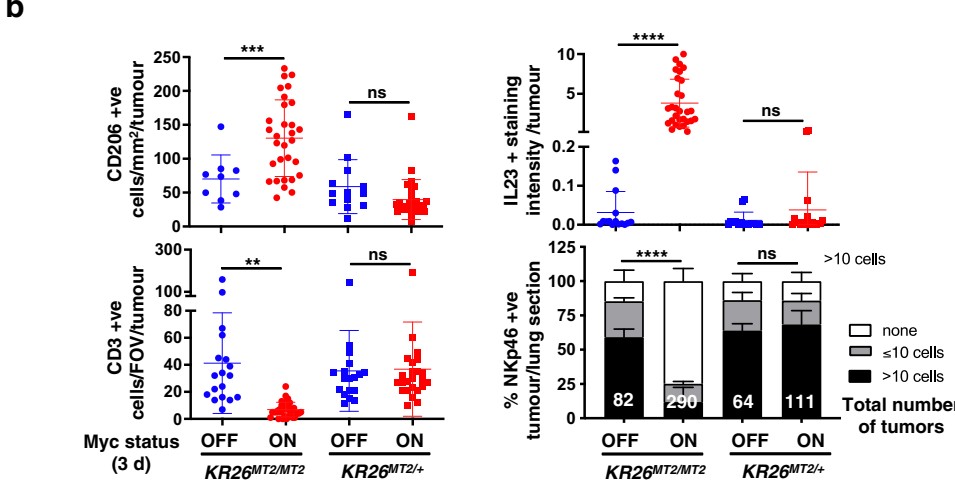

**Fig. 4 | A minimum threshold level of Myc is required to engage instructive stromal signals that drive transition from indolent pre-tumour to invasive neoplasia. a** Representative IHC analysis of in sections of lungs harvested from $KR26^{MT2/MT2}$ and $KR26^{MT2/+}$ mice 12 weeks after activation of KRas$^{G12D}$, either without or with activation of MycER$^{T2}$ for 3 days. IHC staining is shown for CD206$^+$ macrophages (top row), IL23 (second row), CD3$^+$ T cells (third row), and NKp46$^+$ NK cells (bottom row) in sections of lungs harvested from $KR26^{MT2/MT2}$ and $KR26^{MT2/+}$ mice 12 weeks after activation of Kras$^{G12D}$, either without or with activation of MycER$^{T2}$ for 3 days. **b** Quantification of CD206, IL23, CD3 and NKp46 immunohistochemical staining (IHC) in sections of lungs described in A. Results depict mean ± SD of independent tumours from 3-6 mice per treatment group. IL23 + staining intensity was normalised to the number of nuclei per FOV (Field of view) as described in ref. 66. For NKp46 staining, tumours connected to clearly distinguishable vascular and airway regions were considered; the number of tumour-associated NKp46$^+$ cells per tumour per lung section were counted. Data were analysed using unpaired t-test with Welch's correction and two-tailed analysis (CD206, IL23, CD3) or two-way ANOVA (NKp46). For CD206 staining, $n = 9$ independent tumours (3 mice) for Myc

OFF and $n = 30$ independent tumours (6 mice) for Myc ON $KR26^{MT2/MT2}$ groups with $p = 0.0009$; $n = 13$ independent tumours (3 mice) for Myc OFF and $n = 27$ independent tumours (4 mice) for Myc ON $KR26^{MT2/+}$ groups with $p = 0.1427$. For IL23 staining, $n = 15$ independent tumours (3 mice) for Myc OFF and $n = 29$ independent tumours (6 mice) for Myc ON $KR26^{MT2/MT2}$ groups with $p < 0.0001$; $n = 15$ independent tumours (3 mice) for Myc OFF and $n = 20$ independent tumours (4 mice) for Myc ON $KR26^{MT2/+}$ groups with $p = 0.2339$. For CD3 staining, $n = 18$ independent tumours (3 mice) for Myc OFF and $n = 32$ independent tumours (6 mice) for Myc ON $KR26^{MT2/MT2}$ groups with $p = 0.0011$; $n = 18$ independent tumours (3 mice) for Myc OFF and $n = 24$ independent tumours (4 mice) for Myc ON $KR26^{MT2/+}$ groups with $p = 0.8982$. For NKp46 staining, $n = 82$ independent tumours (3 mice) for Myc OFF and $n = 290$ independent tumours (6 mice) for Myc ON $KR26^{MT2/MT2}$ groups with adjusted $p < 0.0001$ for > 10 cells; $n = 64$ independent tumours (3 mice) for Myc OFF and $n = 111$ independent tumours (4 mice) for Myc ON $KR26^{MT2/+}$ groups with adjusted $p = 0.8311$ for >10 cells. **$p < 0.01$, ***$p < 0.001$, ****$p < 0.0001$, ns nonsignificant, SD standard deviation. Source data are provided as a Source Data file.

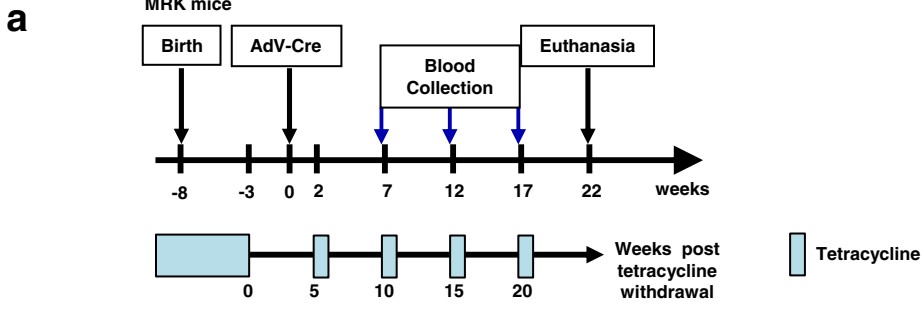

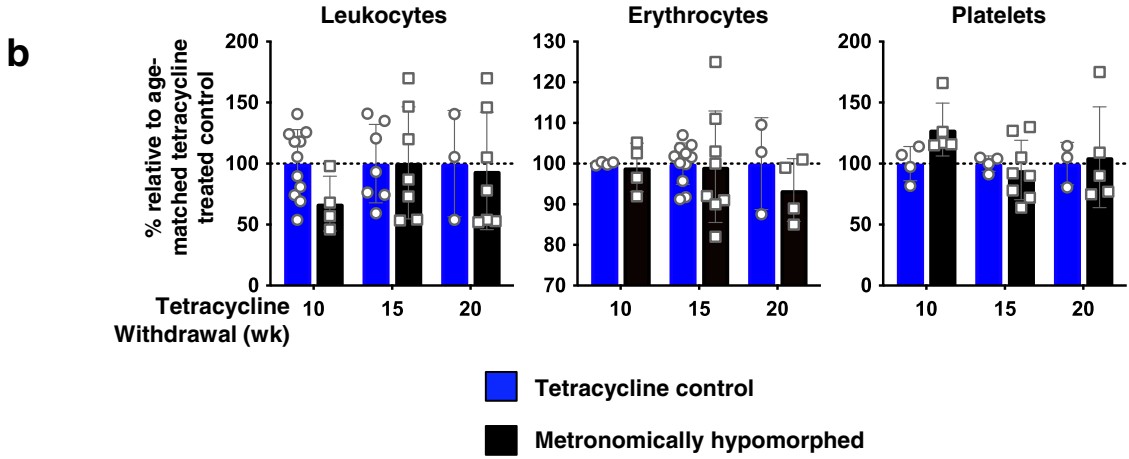

**Tetracycline control**

**Metronomically hypomorphed**

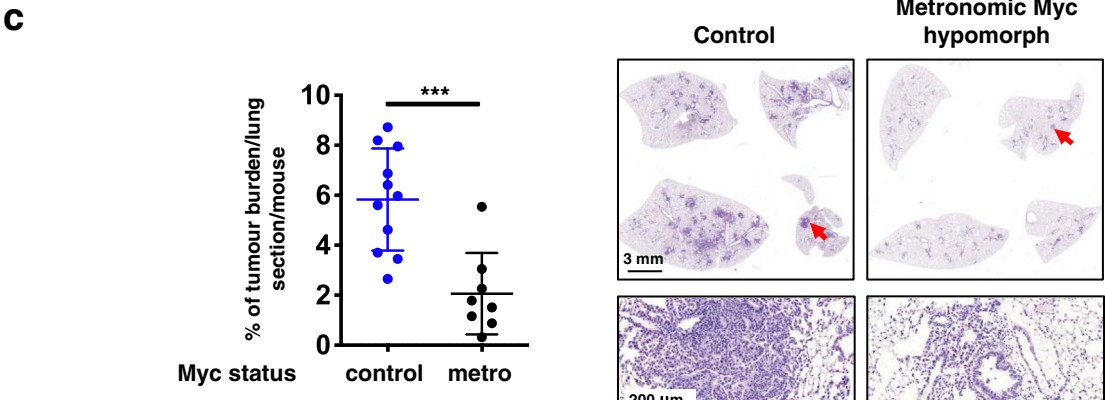

Myc hypomorphism might elicit, we constructed a mouse model in which endogenous Myc expression may be acutely, systemically and reversibly hypomorphed (Supplementary Fig. 1) to a degree broadly concordant with the ~30-60% reduced levels of Myc present in tissues of $Myc^{+/-}$ haploinsufficient mice[30] and mice in which key $Myc$ enhancer elements have been deleted[13]. The model allows us to toggle reversibly the level of endogenous Myc at will, imposing rapid, reversible, and highly consistent hypomorphism of endogenous Myc in all tested tissue.

Imposition of hypomorphism of endogenous Myc expression in adult mice profoundly retarded subsequent development of $KRas^{G12D}$-driven lung and pancreas tumours, reminiscent of the reduced tumour incidence reported for germline $Myc$ hemizygous and enhancer-deleted mice[13,21,23,24,27,28,31]. Thus, activation of $KRas^{G12D}$ in adult lung epithelium of control animals typically drives sporadic development of

multiple invasive adenomas and adenocarcinomas[37,43], covering a spectrum of severities over our 32-week experimental timeframe. By contrast, Myc-hypomorphed mice remained essentially tumour-free for the entire duration with almost all lesions stalling at the stage of focal hyperplasia (AAH) save for a very few low-grade 1 lesions. In this study, Myc hypomorphism was imposed prior to focal activation of $KRas^{G12D}$, from which we inferred that the protection afforded by Myc hypomorphism acted either before or after $KRas^{G12D}$ activation but prior to the transition from hyperplasia to invasive cancer. To narrow down the location of the bottleneck we then used a well-documented pancreatic adenocarcinoma mouse model in which $KRas^{G12D}$ activation is induced by Cre expression driven from the $pdx$ promoter in pancreatic progenitor cells in early embryogenesis (E8)[46], long before Myc hypomorphism is imposed in adult mice. Similarly to lung, the overall impact of Myc hypomorphism was a profound block in the transition

**Fig. 5 | Metronomic imposition of Myc hypomorphism protects against KRas$^{G12D}$-driven lung tumourigenesis without triggering associated haematological pathologies. a** Schematic of metronomic prevention study. Endogenous Myc was maintained in *MRK* mice at normal physiological *wt* levels throughout development and into adult life by continuous administration of tetracycline. At 5 weeks of age tetracycline was withdrawn to hypomorph endogenous Myc and 3 weeks later KRas$^{G12D}$ sporadically activated in lung epithelium by Adv-Cre inhalation. Two weeks later, mice were put on a metronomic regimen of 1 week's relaxation from hypomorphism followed by 4 weeks' re-imposition of Myc hypomorphism, and this was then repeated for a total of 4 cycles. Peripheral blood was withdrawn for analysis just prior to each 1 week's relaxation period. Normal levels of endogenous Myc were maintained in control mice by continuous administration of tetracycline throughout. Mice were euthanized 22 weeks post Adv-Cre activation of *KRas$^{G12D}$* and tissues harvested. **b** Metronomic imposition of Myc hypomorphism spares mice from haematological pathology. Blood counts of total leukocytes, erythrocytes and platelets taken at various times of the metronomic regimen described (10, 15 and 20 weeks) showing substantial amelioration of the pathological changes caused by sustained Myc hypomorphism. Control mice were maintained on tetracycline throughout. Results represent mean ± SD in each treatment group. Multiple Unpaired *t* test with Welch correction, single pooled variance, Holm-Šídák method, shows non-significant difference between groups. For leukocytes, at 10 weeks $n = 11$ and $n = 4$ for non-Myc hypomorphed and metronomically Myc hypomorphed mice, respectively, with adjusted $p = 0.368391$; at 15 weeks $n = 7$ for both groups with adjusted $p = 0.974878$; at 20 weeks $n = 3$ and $n = 7$, respectively, with adjusted $p = 0.966531$. For erythrocytes, at 10 weeks $n = 4$ for both non-Myc hypomorphed and metronomically Myc hypomorphed mice with adjusted $p = 0.978717$; at 15 weeks $n = 11$ and $n = 8$, respectively, with adjusted $p = 0.978717$; at 20 weeks $n = 3$ and $n = 4$, respectively, with adjusted $p = 0.707513$. For Platelets, at 10 weeks $n = 4$ and $n = 5$ for non-Myc hypomorphed and metronomically Myc hypomorphed mice, respectively, with adjusted $p = 0.286349$; at 15 weeks $n = 3$ and $n = 9$, respectively, with adjusted $p = 0.928754$; at 20 weeks $n = 3$ and $n = 5$, respectively, with adjusted $p = 0.928754$. **c** Left: percentage lung tumour burden relative to total lung in control (normal Myc level – blue) versus metronomically Myc hypomorphed *MRK* mice (black) 22 weeks post Adv-Cre activation of *KRas$^{G12D}$*. Results depict mean ± SD in each treatment group. The unpaired t-test Welch's correction and two-tailed analysis was used to analyse tumour burden. ***$p < 0.001$. SD = standard deviation. $n = 11$ for non-Myc hypomorphed and $n = 8$ for metronomically Myc hypomorphed mice with $p = 0.0004$. Right: Representative H&E-stained sections of lungs from control (normal Myc level) versus metronomically Myc hypomorphed *MRK* mice 22 weeks post Adv-Cre activation of *KRas$^{G12D}$*. Arrows mark regions shown at higher magnification below. Source data are provided as a Source Data file.

of almost all indolent PanIN pre-tumour lesions to invasive cancer, temporally locating the Myc-dependent bottleneck post initial KRas activation but before the invasive expansion of early cancer. However, in contrast to the p53-proficient lung model, where essentially all advanced neoplasia was blocked, we did observe occasional large "escapee" tumours in the mutant p53 PDAC model. Closer inspection revealed that virtually all the escapee tumours had "broken" our hypomorphing model, either by significantly upregulating Myc and/or losing expression of the tTs$^{kid}$ repressor. Analogous occasional "escapees" were also observed in a variant of our lung model in which p53 had been deleted at the same time as KRas$^{G12D}$ activation. From this, we conclude that inactivation of p53 does not, of itself, abrogate cancer protection afforded by hypomorphic Myc: however, it greatly increases the chance of genetic accidents wrecking our experimental model. We have previously reported a similarly increased tendency to loss of transgenes and general genetic instability in p53-deficient GEMMs[55]. Of note, we never observed any escapee lung tumours in p53-competent, Myc hypomorphed mice over the entire 32-week duration of our study. This indicates that the rate of sporadic inactivation of p53 in the hypomorphism-stalled lesions, which would presumably have led to sporadic escapee adenocarcinomas, is negligible.

Together, our data indicate that Myc hypomorphism stalls cancer evolution at a critical early bottleneck that temporally coincides with the transition from indolent hyperplasia to locally invasive neoplasia. To search for a causal mechanism by which Myc hypomorphism would block this evolutionary transition we asked the counter question - what is the minimum level of Myc required to drive stalled hyperplasias through the bottleneck? We recently demonstrated that ectopic Myc expressed at quasi-physiological levels is sufficient to drive immediate and synchronous transition of stalled, indolent KRas$^{G12D}$-driven lung AAHs or pancreatic PanINs into, respectively, aggressive and invasive lung and pancreatic adenocarcinomas, along with all the signature stromal sequelae associated with each adenocarcinoma type[37,38]. Remarkably, a mere 50% reduction in Myc expression, to a level comparable with the reduction in endogenous Myc expression in our switchably hypomorphed *MR* mice, proved completely inert in driving the transition from indolent pre-cancer to overt cancer. This same reduced level of ectopic Myc also failed to drive either of the two key instructive signals, CCL-9-mediated chemoattraction of VEGF + PD-L1 + alveolar macrophages and IL-23-mediated exclusion of innate and adaptive lymphoid cells, which we had previously shown to be absolutely required for Myc-dependent invasion and macroscopic expansion of lung AAHs[37]. Thus, hypomorphic levels of Myc are insufficient to engage instructive stromal-remodelling signals required for progression of pre-malignant hyperplasia to frank, invasive cancer.

A comparative analysis in pancreas of Myc target genes induced by bi-allelic *Rosa26* (MycER$^{T2}$ levels analogous to those of *wt* endogenous Myc and competent to drive the transition of indolent KRas$^{G12D}$-driven PanIN to pancreatic adenocarcinoma) versus mono-allelic *Rosa26* (MycERT$^2$ levels equivalent to hypomorphed endogenous Myc and insufficient to drive transition to adenocarcinoma) revealed varying impacts of Myc hypomorphism on Myc target gene expression. Some appeared unaffected by the 50% reduction in Myc level, others were no longer induced at all, and most fell somewhere in between. Further analysis will be needed to establish causal connections relating Myc transcriptional output with hypomorphic phenotypes. Acutely Myc hypomorphed adult animals showed no obvious short-term deleterious impact on normal adult physiology or health and even long-term imposition of Myc hypomorphism caused only mild and rapidly reversible pathologies, principally gradual, relatively mild and completely reversible haematopoietic dysfunction. Since total acute ablation of Myc in adult bone marrow is rapidly fatal[56], the survival of acutely hypomorphed mice therefore attests to the significant residual transcriptional activity that hypomorphic Myc levels retains. However, this begs the question of why evolution should have set endogenous Myc expression at levels that appear both "unnecessarily" high in most tissues and pro-neoplastic to boot. To address this conundrum, we used the temporal control afforded by our switchable hypomorph model to search for physiological processes that are sensitive to Myc hypomorphism. Myc expression in mouse embryos and their placentas is essential for successful transit through mid-gestation[26,57], a highly proliferative stage at which widespread complementary co-expression of the Myc paralogue N-Myc subsides[58] and subsequent tissue growth and maturation consequently switches to sole dependence on Myc. *Myc$^{-/-}$* embryos die around 10-11 *dpc* due to widespread organogenesis failure, most notably in vasculogenesis and haematopoiesis[49]. Similarly, sustained imposition of Myc hypomorphism from conception in *MR* mice induced fully penetrant embryo failure. This occurred rather later than in *Myc$^{-/-}$* embryos but shares its lethal hypovascular phenotype. Intriguingly, this ubiquitous mid-term lethality of actively hypomorphed *MR* embryos is not evident in germline haploinsufficient *Myc$^{+/-}$* mice, almost all of which survive to adulthood. The reason for this difference is unclear but it underscores the dramatically different phenotypic consequences arising from passive germline *Myc* gene insufficiency versus our active suppression of *Myc* expression in adult animals. Unexpectedly, the specific mid-gestation failure of Myc hypomorphed *MR* embryos was observed only

when the tTS$^{Kid}$ repressor was contributed by the fathers: when mothers contributed the repressor, and were therefore themselves hypomorphed during pregnancy, no offspring of either *MR* or control genotypes were recoverable. Female infertility had also been reported in *Myc$^{+/-}$* haploinsufficient mice, although with markedly less penetrance, and we guess indicates acute dependence on a minimal threshold of Myc expression for some early maternal function such as implantation, decidualization, or endometrial proliferation or angiogenesis[51]. Notwithstanding the proximal cause of infertility of hypomorphed *MR* mothers, however, reversal of Myc hypomorphism rapidly returned all females to full fertility.

In addition to the intense proliferative burst in mid gestation, similarly high levels of proliferation continue in many organs during later gestation and post-natal growth. Nonetheless, imposition of sustained Myc hypomorphism in either *MR* embryos post 13.5 *dpc* or in neonatal *MR* mice had no discernible impact on any aspect of development or on proliferation in any developing organ. Pups that were continuously hypomorphed from birth developed normally, achieving normal adult size and weight at the same time as their control littermates.

Although the iatrogenic consequences of short-term partial Myc inhibition appear surprisingly few and confined principally to mid-term gestation and female fertility, cancer prophylaxis would presumably require systemic Myc inhibition over a large fraction of adult life. Therefore, to model this we subjected adult *MR* mice to extended Myc hypomorphism. This uncovered a delayed but significant impact on haematopoiesis, starting with progressive onset leukopenia from around 6 weeks and consistent with the known obligate role of Myc in adult haematopoiesis, in particular in the maturation of haematopoietic stem cells[56,59]. From around 12 weeks mice exhibit mild erythropenia, followed later by thrombocytosis (18 weeks), both consistent with an established role Myc plays in promoting erythropoiesis over megakaryocytopoiesis[60]. These blood changes were accompanied by the development of modest splenic extramedullary haematopoiesis in around half the animals. However, all these haematopoietic pathologies rapidly reversed upon relaxation of Myc hypomorphism and, on this basis, we subjected *MR* mice to repeated metronomic Myc hypomorphism (4 weeks Myc suppression interspersed with 1-week recovery). This completely abrogated all the haematopoietic pathologies yet remained fully effective at blocking progression of KRas$^{G12D}$-driven lung tumours.

The extent to which pharmacological partial Myc suppression in humans would replicate the cancer prophylaxis we see in mice awaits clarification because no specific pharmacological Myc inhibitor has yet been devised. Nonetheless, valid pharmacological strategies for indirectly inhibiting Myc already exist, although all such current agents are relatively non-specific, targeting Myc transcription, translation or stability[61], and typically suppressing Myc only transiently and partially, and with significant off-target impacts. While this has greatly limited their therapeutic utility in cancer treatment, where sustained and complete Myc inhibition is sought, our data indicate that even the partial and episodic inhibition of Myc afforded by such indirect agents might yet prove effective for cancer prophylaxis and that a tolerable regimen might perhaps be found that balances prophylactic efficacy with minimal side effects.

## Methods

### Generation and maintenance of genetically engineered mice

*p53$^{flox}$; LSL-Kras$^{G12D}$, Pdx-1-Cre, LSL-p53$^{R172H}$, Rosa26-LSL-MER$^{T2}$* and *β-actin-tTS$^{Kid}$* mice have all been previously described[40,43,48,62–64]. To generate of *Myc$^{TRE}$* mice, a targeting vector was constructed (Supplementary Fig. 1) to include a heptameric tetracycline-response element (*TRE*) derived from the pTRE2 vector (Clontech), inserted into the intron 2 of the endogenous Myc locus that serves as a binding site for tetracycline-regulatable transcriptional repressors; silent mutations

incorporated into exon3 (exon3 WT-GAAGAAGAG-; exon3 Mut-GAG-GAGGAA-) to distinguish it from endogenous exon 3 of Myc, *LoxP*-flanked Neomycin resistance cassette L2 Neomycin used for positive selection; and a HSV-TK cassette added at the 3′end of the targeting vector to allow negative selection in mouse embryonic cells. Mouse embryonic stem cells (mESCs), from which *Myc$^{TRE}$* mice were derived, were produced via homologous recombination of the endogenous Myc locus. All mice were all backcrossed to *C57BL6J* background. For experimental purposes, mice were age-matched and each experimental group included both sexes (females and males).

Mice were maintained on a 12-h light/dark cycle with continuous access to food and water. The environmental conditions were as follows: temperature 20–24 °C (68–75 °F) and humidity 55% ± 10%. All animal studies were reviewed and approved by the Animal Welfare and Ethical Review Body (AWERB) of the Cancer Research UK Cambridge Institute and the Francis Crick Institute and licensed by the UK Home Office (license number PP2645677 and protocols numbers 2 and 3.) The ARRIVE guidelines were adhered to throughout. For Adenovirus-Cre recombinase (AdV-Cre) delivery, 8-10 week-old mice were anaesthetised with isoflurane (Zoetis, IsoFlo 250 ml) and $5 \times 10^7$ p.f.u. (plaque-forming units) of AdV-Cre were administered as described previously[65]. Deregulated Myc activity was engaged in lung epithelia of *KM* mice or pancreatic epithelia of *KM$^{pdx1}$* mice by daily intraperitoneal (i.p.) administration of tamoxifen (Sigma-Aldrich, TS648) dissolved in sunflower oil at a dose of 1 mg/20 g body mass; sunflower oil carrier was administered to control mice. In vivo, tamoxifen is metabolised by the liver into its active ligand 4-Hydroxytamoxifen (4-OHT). For long-term treatment, mice were placed on tamoxifen diet (Harlan Laboratories UK, TAM400 diet); control mice were maintained on regular diet. For Myc hypomorphism studies, *Myc$^{TRE/TRE}$* mice were crossed into the *β-actin-tTS$^{Kid}$* mice that ubiquitously express a chimeric transcriptional repressor molecule (tTS$^{Kid}$) that can reversibly inhibit the expression of Myc endogenous gene. To keep systemic Myc at normal physiological level, *Myc$^{TRE/TRE}$; β-actin-tTS$^{Kid}$/–* (*MR*) mice were maintained on drinking water containing tetracycline hydrochloride (100 mg/L) (Sigma-Aldrich, T7660) and 3% sucrose (Sigma-Aldrich, S9378) to increase palatability. To induce Myc hypomorphism, *MR* mice were transferred onto 3% sucrose drinking water.

### Embryo isolation

Embryos from timed MR matings (♀$M^{TRE/TRE}$ × ♂$M^{TRE/TRE}$;$R^{+/-}$) were isolated, dissected and rinsed with PBS at 4 °C. Images were acquired using an M50 Leica Stereo microscope.

### Tissue preparation and histology

Tissues were isolated, fixed overnight in 10% neutral-buffered formalin, (Sigma-Aldrich, 501320), stored in 70% ethanol and paraffin embedded. Tissue sections (4 μM) were stained with hematoxylin and eosin (H&E) using standard reagents and protocols. Peripheral blood (50–80 μl) from the tail vein was collected in K$_2$EDTA-coated BD Microtainer™ tubes (Fisher, 10346134) and analysed at the Central Diagnostic Services, Queen's Vet School Hospital, University of Cambridge.

### Immunohistochemistry and RNA scope

For immunohistochemistry (IHC), paraffin-embedded sections were de-paraffinized, rehydrated, and either boiled in 10 mM citrate buffer (pH 6.0) for Ki67, CD206, CD3, and IL23 staining, in Tris/EDTA buffer (pH8.0) for Myc staining or proteinase K treatment for NKp46 staining. Primary antibodies used were as follows: rabbit monoclonal anti-Ki67 (clone SP6, 1:100 dilution) (Lab Vision, Fisher, 12603707), goat polyclonal anti-CD206 antibody (1:200 dilution, R&D Systems, AF2535), rabbit monoclonal IL-23 (EPR5585(N), 1:100 dilution) (Abcam, ab190356), rabbit monoclonal anti-CD3 (Clone SP7, undiluted) (Fisher Scientific, RM9107RQ), mouse monoclonal anti-NKp46 (clone 29A1.4, 1:100 dilution) (Biolegend, 137601) and rabbit monoclonal anti-Myc

(Y69, 1:500 dil) (Abcam, ab32072). Primary antibodies were incubated with sections overnight at 4 °C and detected using Vectastain Elite ABC HRP Kits (Vector Laboratories: Peroxidase Rabbit IgG PK-6101) and DAB substrate (Vector Laboratories, SK-4100); slides were then counterstained with hematoxylin solution (Sigma-Aldrich, GHS232). RNA-in situ hybridization (RNA-ISH) for *tTS^Kid* was performed with a custom-designed probe, amplified with RNAscope 2.5 HD Reagent Kit (Advanced Cell Diagnostics; 322300) and developed with the TSA Plus kit (PerkinElmer; NEL760001KT) according to manufacturer's instructions. Images were collected with a Zeiss Axio Imager M2 microscope equipped with Axiovision Rel 4.8 software.

### Immunoblotting analysis of mouse adult lung fibroblasts
Lungs from 5- to 8-week-old mice were dissociated in Hank's Balanced Salt Solution containing 100 mg/mL type II collagenase (Fisher Scientific 10738473) and passed through 70 µm filters to obtain single-cell suspensions. Mouse adult lung fibroblasts (MALFs) were cultured and maintained in DMEM (Thermo Fisher 41966052) supplemented with penicillin-streptomycin (Thermo Fisher, 15140-122), L-glutamine (Thermo Fisher, 25030-024), and FBS (Scientific labs, F7524). For experimental purposes, MALFs were treated with 1 µg/ml doxycycline hyclate (Sigma D9891). Proteins from MALFs were extracted using standard protocols. Total protein lysates from MALFs were electrophoresed on an SDS-PAGE gel and blotted onto Immobilon-P membrane (Millipore). Membranes were blocked with 5% nonfat milk and primary antibodies incubated overnight at 4 °C. Secondary antibodies were applied for 1 h followed by chemiluminescence visualisation. The following primary antibodies were used: MYC (ab32072; Abcam, 1:2000 dilution) and β-actin (sc-69879, Santa-Cruz Biotechnology, 1:5000 dilution). HRP-conjugated secondary antibodies: goat anti-rabbit (sc-2301; Santa-Cruz Biotechnology 1:7500 dilution) and goat anti-mouse (A4416, Sigma-Aldrich, 1:7500 dilution).

### RNA-sequence analysis
Total RNA was isolated from snap-frozen (liquid nitrogen) *KCR26M^{MT2/MT2}* and *KCR26M^{MT2/+}* mouse pancreas using a PureLink RNA Mini kit (ThermoFisher scientific) and PureLink DNASe set (ThermoFisher scientific). The quality, quantity and integrity of RNA were assessed by NanoDrop1000 spectrophotometer and 2100 Bioanalyzer (Agilent Technologies, CA). RNA libraries were generated using TruSeq Stranded mRNA Library Prep kit following manufacturer's instructions (Illumina, CA). RNA Libraries were run on an Ilumina NextSeq 500 using the 75-cycle high-output kit (single-end sequencing) at the Cambridge Genomic Services (CGS) at the University of Cambridge (https://www.cgs.path.cam.ac.uk). The sequencing data was analysed by the bioinformatics team and quality checked using FastQC v0.11.4. In summary, reads were trimmed using TrimGalore v0.5.0 and those less than 20 bases long discarded. Reads were mapped using STAR v2.7.1. The Ensembl *Mus Musculus* GRCm39 (release 103) reference genome was used with annotated transcripts from the Ensembl *Mus Musculus*. GRCm39.103.gtf file. The number of reads mapping to genomic features was calculated using HTSeq v0.6.1. Differential Gene Expression Analysis using the counted reads employed the R package edgeR v3.26.5 and the unpaired generalised linear model (glm) as suggested in edgeR user's guide. The data has been bias corrected using the CQN package (version 1.24.0) within the model. RNA-seq data have been deposited in the ArrayExpress database at EMBL-EBI (www.ebi.ac.uk/arrayexpress) under accession number E-MTAB-10807 (https://www.ebi.ac.uk/arrayexpress/experiments/E-MTAB-10807/).

### Quantitative real time-PCR
Tissues were collected at appropriate time points and snap-frozen in liquid nitrogen. Total RNA was isolated using a Qiagen RNeasy Plus Isolation Kit followed by cDNA synthesis (High Capacity cDNA RT kit, Applied Biosystems, 4374966). RT-PCR was performed using TaqMan Universal Master Mix II (Fisher, 4440038), according to manufacturer's protocol. Primers used were: *Myc* (Fisher, Mm00487804_m1) and *Tbp* (Fisher, Mm00446973_m1). Samples were analysed in triplicate on an Eppendorf Mastercycler Realpex 2 with with Eppendorf Quantstudio design & Statistical Analysis Vl.2 Software. *Tbp* was used as an internal amplification control.

### Quantification and statistical analysis
For quantification of tumour burden, H&E sections were scanned with an Aperio AT2 microscope (Leica Biosystems) at 20X magnification (resolution 0.5 microns per pixel) and analysed with Aperio Software v12.1.0.502. Generally, one mid-section per mouse was analysed; if in doubt, consecutive slides were analysed to distinguish between individual tumours. IHC stained slides were quantified using Fiji open-source software. Unless otherwise specified, statistical significance was assessed by unpaired t-test with Welch's correction and two-tailed analysis, with the mean values and the standard deviation (SD) calculated for each group using Prism GraphPad software. Unpaired *t*-test with Welch's correction and two-tailed analysis was also employed in RT-PCR statistical analysis. *p* values (ns = non-significant; \**p* < 0.05, \*\**p* < 0.01, \*\*\**p* < 0.001, \*\*\*\**p* < 0.0001) for each group.

### Reporting summary
Further information on research design is available in the Nature Research Reporting Summary linked to this article.

### Data availability
The RNA-seq data generated and analysed in this study have been deposited in the ArrayExpress database at EMBL-EBI (www.ebi.ac.uk/arrayexpress) under accession number E-MTAB-10807 (https://www.ebi.ac.uk/arrayexpress/experiments/E-MTAB-10807/). All data that support the findings of this study are included in the supplementary information files. The raw data are available in the Source Data file whenever possible. Source data are provided with this paper.

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

## Acknowledgements

We thank the members of the Evan laboratory for invaluable discussion and advice. We also thank Stephanie Whike, Michaela Griffin and Deborah Breiner for assistance with histology and genotyping. The study was supported by Cancer Research UK programme grants (C4750/A12077, C4750/A19013A, and C4750/A29210), a European Research Council Advanced Investigator Award (294851), and a Stand Up To Cancer-Cancer Research UK-Lustgarten Foundation Pancreatic Cancer Dream Team Research Grant (SU2C-AACR-DT20-16), all to G.I.E. Stand Up To Cancer is a division of the Entertainment Industry Foundation and the research grant is administered by the American Association for Cancer Research, the Scientific Partner of SU2C.

## Author contributions

N.M.S. and G.I.E. conceived the project. G.I.E. supervised the study with help from L.B.S. and T.D.L.; N.M.S. designed experiments with help from L.P.; N.M.S. and L.P. performed all experiments. R.M.K., Y.W.K., T.C., S.K., D.G., A.P., P.A. and M.J.A. assisted with some experiments. N.M.S. and G.I.E. wrote the manuscript. T.D.L. edited the manuscript. All authors discussed results and revised the manuscript.

## Competing interests

G.I.E. is a member of AstraZeneca's IMED oncology external science advisory panel. The remaining authors declare no competing interests.

## Additional information

**Correspondence and requests** for materials should be addressed to Nicole M. Sodir or Gerard I. Evan.

