## [Peer Review File · Nature Communications]

REVIEWER COMMENTS

Reviewer #1 (Remarks to the Author):

Sodir et al. analyze a novel model of reversible c-Myc hypomorphism to address quantitative aspects of Myc physiology and to acquire information that could guide the potential of drugs that might inhibit MYC activity as a strategy for cancer prevention. Needless to say, this is an important area and the approach taken is innovative and very valuable, harnessing several genetic mouse models - published and newly developed here. The work shows that the levels of c-Myc are a bottleneck for early stages of tumor development, that a 50% reduction of MYC expression is likely to impact on the homeostasis of tissues with constitutive renewal and that a metronomic reduction of c-Myc levels might be an adequate strategy to prevent malignant transformation while maintaining tissue homeostasis. Some of the results suggest a potential interaction between c-Myc hypomorphism and Trp53 mutations; this aspect might merit additional investigation, particularly given the role of Trp53 inactivation in the occurrence of tumor escapees. Overall, the conclusions are substantiated by the data. However, the results reported are rather descriptive and superficial and a more in-depth analysis would significantly strengthen the quality of the manuscript.

Specific comments

1. The number of mice used in many experiments is very low.
2. Quantification of histological and immunohistochemical images is poorly described: how many sections from independent regions of the block, how many fields, how were fields selected, details of the statistical analysis.
3. The analysis of the effects of hypomorphism on normal tissue homeostasis (no stress) are somewhat limited (Suppl Figure 2): Ki67 expression in the intestine and CFU assays in the bone marrow would provide a more convincing evidence on the normality of the hypomorphic mice. I suggest to include the data from Supplementary Figure 3 here, rather than later in the paper. A more refined molecular analysis would be worthwhile as fine alterations would not be detected at the histological level; RNA-Seq would be a valuable strategy to identify more subtle molecular changes. This is suggested by the more pronounced phenotype at time points later than 2 weeks (page 12).
4. Figure 1B: histological analysis of lesions should include quantification of hyperplastic lesions vs. adenomas vs. adenocarcinomas. More details required in order to dissect effect on initiation vs. progression.
5. Figure 2A: histological analysis of lesions should include quantification of low- and high-grade PanINs. It is striking that the pancreas of hypomorphed mice seems to contain mucinous PanINs since the KPC model rarely develops these lesions (unlike the KP model), suggesting an interaction with mutant p53. This merits some mechanistic analysis, especially considering the escapee tumors. Analysis of mutant p53 expression in the hypomorph mice would be interesting to understand the stalling of PanIN progression.
6. Figure 2 and others: The quantification using as only parameter the %Myc+ve cells is somewhat superficial.
7. Figure 3. The fact that this lung cancer model is based on the Trp53 lox/lox alleles rather than in the mutant allele used in KPC mice renders the comparison with the pancreatic cancer model complex. The number of samples analyzed in panel 3B is very low.
8. Figure 4. Validation of quantification of MycER expression in the current series of mice would be valuable, rather than relying on previous work. The "Tumor burden" parameter used for quantification is very crude; a finer assessment is important. It is proposed that CCL9 and IL-23 are required for the macrophage influx and lymphocyte depletion but neither of them is measured and there is no functional data allowing to conclude on the effects.

9. The transcriptome analysis shown in Supplementary Figure 5 is fragmentary and selective. A full analysis of the RNA-Seq data should be provided.

Minor comments

1. The precise duration of "long term TMX diet" should be indicated since it is known that after long periods of administration can result in gastrointestinal damage.
2. The unpaired t-test is not appropriate unless the data show a normal distribution, which requires a larger sample size. Therefore, Mann-Whitney should be used, instead.
3. Review carefully for typos.

Reviewer #2 (Remarks to the Author):

The manuscript by Sodir et al, "Reversible Myc hypomorphism identifies a key Myc-dependency in early cancer evolution" explores the roles of elevated versus sub-physiological (hypomorphic) levels of MYC in tumorigenesis and in tumor progression. In this work the authors use a variety of approaches to vary murine MYC levels from low to high levels in KRasG12D-driven lung and pancreatic tumors and followed the natural history of the resulting pre-neoplastic hyperplasia, to in situ tumors on to full-blown malignancy. For most of the experiments, the investigators used an endogenous MYC-allele engineered to be hypo-morphable through tet-off regulated repression using a KRAB-domain-tet repressor fusion recruited to MYC intron 2. For other experiments, they returned to the use of the rosa-MYC-ERT system that they have extensively characterized in prior studies. The authors report that enforced hypomorphic levels of MYC prevent or delay lung tumorigenesis and they show that the bottleneck to tumor formation correlates with reduced expression of genes such as Il23 and CCL9 that are involved in tumor invasion and they report an attenuated stromal response. They also show that the salutary influence of hypomorphic MYC is not a simply the result of the execution of high-MYC tumors by p53, because protection by low MYC persists in the complete absence of p53. The authors study the effect of doxycycline-imposed hypomorphic MYC on prenatal development and postnatal maturation and growth. They make several interesting observations about the lowering of MYC levels in their system. They observe that very early gestation is affected more for all embryos if the active KRAB-tetR is maternally inherited than when transmitted through sperm, likely due to the uterine competence. With early reduction of MYC the hypomorphic embryos die by E13.5, after that development proceeds apparently normally despite low levels of MYC. Post-natally, hypomorphic MYC mice sustained by doxycycline-withdrawal, eventually became anemic, but such anemia could be prevented with periodic short pulses of doxycycline. The investigators show that the protective effect of MYC-reduction on tumors endured in the face of these periodic bone-marrow sparing pulses.

This is an interesting study and potentially important study that may help to instruct the development of anti-MYC therapy. However, some of these experiments are incremental advances over existing studies, and for some of the more original and creative parts of the manuscript, some interesting experiments remain to be performed or at least considered. In addition important papers in the literature need to be considered here. The authors need to stress more the experiments, conclusions and implications that are new versus those that are derivative.

Major points:

1. The system for hypomorphing MYC is interesting and potentially a very valuable tool. But there are other studies that need to be cited that show that in experimentally-engineered systems, MYC-haploinsufficiency decreases tumors (for example doi: 10.1101/cshperspect.a014290., Fig. 2) Yet, there are also uncited studies (doi: 10.1016/j.cell.2014.12.016..) that show that haploinsufficiency

of MYC does not greatly reduce the incidence of spontaneously arising tumors, nor does this genetic haploinsufficiency provoke any significant pathology during development and maturation. Is there not only a threshold for tumorigenesis, but a threshold for MYC-hypomorphism? If hypomorphic MYC is maintained at haploinsufficient levels, is it possible to protect from both tumors and bone marrow failure? The authors need to compare or at least discuss hypomorphic versus haploinsufficiency of MYC.

2. Multiple papers—starting with Murphy, et al. (also doi: 10.1158/0008-5472.CAN-07-6192) have already suggested that a MYC-threshold must be exceeded for experimentally induced tumors. The MycERT experiments add little new to this study.

3. The levels of MYC in the hypomorphic system are evaluated with and without doxycycline by immunostaining and qPCR, but are never compared to the levels of MYC in wild-type or haploinsufficient animals. With histologic sections, such comparisons always are subject to caveats concerning the relative contributions of different cell types in the selected samples and fields. It is important to compare the MR and wild-type mice to establish that the M allele is not itself somewhat hypomorphic. What is the output of the MR or M alleles compared with the native MYC allele? This is important because the existing literature indicates that MYC^{+/-} yields a relatively mild phenotype in most cases whereas MR mice exhibit more pronounced deficits in development and in hematopoiesis.

4. In this light it would be good to evaluate whether MR/+ mice are tumor prone in the KRAS system. If there is a threshold for MYC in tumorigenesis, would MR/+ be above or below that threshold (MR/MR is clearly below it)? Also, use of homozygous MR mice protects from the usual genetic events that upregulate MYC in cancer. It would be interesting to observe in heterozygotes, how frequently and how efficiently genetic events up-regulate the wild-type MYC allele to bypass the MR system.

5. It is hard to understand why the RosaMycERT/+ are "hypomorphic" with tamoxifen—don't these mice also have two endogenous wild-type Myc alleles? Expression of RosaMycERT may be low relative to the endogenous allele, but how can the total be hypomorphic? Does this suggest that MYC doesn't need to be hypomorphic, just not elevated to protect from tumorigenesis?

6. It seems that escapees of the MR system lose the repressor. Are there other genetic events that bypass the hypomorphic system?

7. Does the inability of the MR- KRasG12D hyperplastic foci to invade or provoke a stromal response represent the lack of induction of a specific program by MYC or a general transcriptional downregulation lacking sufficient MYC?

Reviewer #3 (Remarks to the Author):

This paper presents a thorough and compelling analysis of the role that MYC expression levels play in tumor progression in mutant KRAS models of lung and pancreatic carcinoma. By cleverly deploying a tetracycline response element in a Myc intron, the authors are able to partially suppress endogenous Myc RNA expression by 20-50%, an effect readily reversible by addition of tetracycline. Their findings are striking in demonstrating that the decreased levels of Myc block the transition from hyperplasia to adenocarcinoma in both the lung and pancreas models. They go on to show that p53 is not required for the block and that "escaper tumors" tend to reestablish high levels of Myc, thus reinforcing the notion that full tumor growth is dependent on sustained higher Myc levels. Indeed, they show that metronomic up-down expression of Myc is sufficient to block tumorigenesis. Moreover, they provide experimental data implicating a failure of the Myc "hypomorphed" tumor to secrete cytokines likely to be involved in reconfiguring the stroma as required for progression.

Beyond the demonstration that Myc levels matter in tumor progression, the relevance of this paper

is that it reinforces the concept of partial suppression of Myc expression as a potential therapeutic route. As the authors point out, the hypomorphed mice do not suffer from major defects in development or growth. Therefore, if we could only figure out how to dial Myc down at an early stage in cancer progression we would have an effective, non-toxic, co-therapy.

Overall, I found the data presented to be very clear and convincing. My only quibble is a lack of quantitation relating to MYC protein levels. While several figures show IHC for MYC (e.g., Fig 2B) the difference in protein levels appeared rather marginal compared to the ISH. Perhaps the authors should consider including immunoblots from the relevant tissues.

Rebuttal Documents

Detailed response to reviewers' comments.

Reviewer 1

Sodir et al. analyze a novel model of reversible c-Myc hypomorphism to address quantitative aspects of Myc physiology and to acquire information that could guide the potential of drugs that might inhibit MYC activity as a strategy for cancer prevention. Needless to say, this is an important area and the approach taken is innovative and very valuable, harnessing several genetic mouse models - published and newly developed here. The work shows that the levels of c-Myc are a bottleneck for early stages of tumor development, that a 50% reduction of MYC expression is likely to impact on the homeostasis of tissues with constitutive renewal and that a metronomic reduction of c-Myc levels might be an adequate strategy to prevent malignant transformation while maintaining tissue homeostasis. Some of the results suggest a potential interaction between c-Myc hypomorphism and Trp53 mutations; this aspect might merit additional investigation, particularly given the role of Trp53 inactivation in the occurrence of tumor escapees.

Overall, the conclusions are substantiated by the data. However, the results reported are rather descriptive and superficial and a more in-depth analysis would significantly strengthen the quality of the manuscript.

We thank the reviewer for her/his comments and agreement that our study is innovative, valuable, and the conclusions substantiated by the data we present. In reference to comments that the results are rather descriptive and superficial, we now provide substantial additional data that directly address the mechanism that impedes the capacity of hypomorphic Myc to drive the transition to adenocarcinoma (i.e. failure of hypomorphic levels of Myc to drive instructive paracrine signals that drive the transition from indolent pre-tumour to locally invasive cancer) by driving local stromal changes (namely, CD206⁺ macrophage influx, IL-23 induction, and exclusion of CD3⁺ T and NKp46⁺ NK cells – see new Figure 4). We also add, clarification of how we performed quantitation of histopathological data, further data showing that insertion of the *TRE* element in the endogenous *Myc* gene has no impact on Myc expression or kinetics of its induction by serum mitogens (Rebuttal Figure 3), and a more comprehensive exposition of transcriptomic data indicating the impact of Myc hypomorphism on Myc transcriptional output (Supplementary Figure 5 and Rebuttal Figure 1).

We respectfully disagree with the reviewer's inference that our data suggest some sort of interaction between Myc hypomorphism and mutant Trp53. We think this is a misunderstanding of our mutant p53 studies. Broadly speaking, our data indicate that absence/mutation of p53 does not in any way bypass the block to tumour progression emplaced by hypomorphic Myc: instead, absence of p53 increases the chances of some stray genetic events breaking our mouse model. We then provide evidence in supplementary figure 4 of the two principal mechanisms of breakage – either through increasing the risk of silencing expression of the tTS^{Kid} repressor transgene (so nothing is hypomorphed any longer) or through upregulation of Myc (in which case Myc is no longer hypomorphed).

Specific comments

1. The number of mice used in many experiments is very low.

All the models we use – sporadic activation of KRas^{G12D} by Adv-CRE inhalation in p53-competent mice, sporadic activation of KRas^{G12D} by Adv-CRE inhalation together with p53 inactivation, the KPC PDAC mouse – generate tens to hundreds of independent neoplastic foci in each mouse lung/pancreas. Each of these foci is an independent tumourigenic event that thereafter independently and sporadically evolves along its own neoplastic trajectory. The independence of such foci and the tumours into which they sporadically progress has been directly ascertained in the *LSL-KRas^{G12D}* lung model ¹, where it is easier to observe discrete probably due to the inhalation route of Cre delivery foci (upon progression, the foci in the pancreas model rapidly spread and tend to merge, making it difficult to distinguish the outlined of distinct tumours). The independence of these pre-tumour KRas^{G12D}-driven lung and pancreas foci is also inherently evident from their subsequent sporadic progression (Figure 1) as well as from the sporadic manner in which they lose expression of the tTS^{Kid} repressor and/or amplify Myc expression in the p53-deficient variant models (Supplementary Figure 4). As such, each mouse in our studies is a host to tens, sometimes hundreds, of independent tumourigenic events so, while numbers of animals may be small (and purposefully kept that way for ethical reasons) our data are highly significant statistically. Although all of our analyses and quantifications were conducted on individual pre-tumour lesions in the original version of this manuscript, we nonetheless depicted the results as averages over each mouse, which was misleading. To illustrate this, we provide for the reviewers Rebuttal Figure 2, which co-plots both the status of individual lesions (small circles) and the averages across all lesions counted in each mouse (large circles). Both are statistically highly significant but given the independent nature of each individual lesion we now present our data lesion-by-lesion in the revised Figure 4. We submit that the statistically unambiguous data we have generated in this lesion-by-lesion analysis obviates any requirement to expand each of the studies using more mice (although we modestly increased the number of mice where possible), which would be deemed ethically unnecessary and would in addition add at least a year to the time taken for resubmission – probably longer in the current post-COVID climate.

2. Quantification of histological and immunohistochemical images is poorly described: how many sections from independent regions of the block, how many fields, how were fields selected, details of the statistical analysis.

We have now added few sentences in the materials and methods section and figure legends to clarify that. Generally, one mid-section per mouse is analyzed; consecutive slides are analyzed when necessary to distinguish between individual tumours. The fields were selected randomly. As mentioned above, the models we use in this manuscript generate tens to hundreds of independent neoplastic foci in each mouse lung/pancreas; at least 4 independent foci per mouse were quantified. Since foci tend to merge in the pancreas model, they were randomly picked from 4 different areas. t-test with Welch's correction was also employed statistical analysis unless stated otherwise.

3. The analysis of the effects of hypomorphism on normal tissue homeostasis (no stress) are somewhat limited (Suppl Figure 2): Ki67 expression in the intestine and CFU assays in the bone marrow would provide a more convincing evidence on the normality of the hypomorphic mice. I suggest to include the data from Supplementary Figure 3 here, rather than later in the paper. A more refined molecular analysis would be worthwhile as fine alterations would not be detected at the histological level; RNA-Seq would be a valuable strategy to identify more subtle molecular changes. This is suggested by the more pronounced phenotype at time points later than 2 weeks (page 12).

We understand the reviewer's interest in establishing just how normal are Myc-hypomorphed mice – it's a topic of some discussion in our laboratory. However, the primary focus of the manuscript is the discovery of a Myc level-dependent bottleneck in early tumour evolution – a bottleneck whereby hypomorphic levels of Myc are unable to drive instructive stromal remodelling and immune-suppressing signals that we have previously showed to be essential for the transition of KRas^{G12D}-driven epithelial pre-tumour cells to invasive adenocarcinoma.

We of course accept that a provocative implication of our discovery is that long-term partial blunting of Myc activity in adults, if achievable by some pharmacological means in the future, might be effective in cancer prophylaxis. For this reason, we have investigated the most obvious potential phenotypes expected of Myc hypomorphic mice. We discovered an embryonic lethality phenotype, a reversible fertility phenotype, and a mild long-term haematopoietic deficit that can be completely mitigated by imposing Myc hypomorphism metronomically – all observations not replicated by Myc haploinsufficiency. We also show that long-term Myc-hypomorphed animals exhibit no significant weight loss, arguing strongly against GI tract dysfunction, and a very different outcome from the lethal impact of total Myc ablation on intestine function and integrity². Likewise, the extramedullary haematopoiesis (a very sensitive indicator of proliferative stress on bone marrow) and leukopenia we see in adult mice subjected to long-term Myc hypomorphism is mild, transient and, moreover, completely abrogated by applying Myc hypomorphism metronomically while retaining resistance to cancer. This is in stark contrast to the catastrophic bone marrow crash in animals in which Myc is completely ablated^{3,4}, and affirming the essential role that Myc nonetheless plays in haematopoiesis. We also show that sustained global imposition of Myc hypomorphism imposed after mid-gestation (E13.5) has no discernible impact on embryonic or neonatal growth to adulthood, nor any discernible deleterious impact on post-natal growth or development, with animals achieving normal weight and organ morphology on time.

We do now include that “more refined” transcriptomic analysis, comparing the transcriptomes induced by Myc at levels able to drive oncogenesis versus 50% reduction in Myc, which has no measurable oncogenic activity (New Supplementary Figure 5). This analysis was conducted in the pancreata of our *KCR26^{MT2/+}* and *KCR26^{MT2/MT2}* models and directly compares the transcriptional output at t+12 hours after Myc activation in pancreatic epithelium at physiological levels (*KCR26^{MT2/MT2}*) that drives immediate transition from PanIN to PDAC, versus 50% reduced Myc level (*KCR26^{MT2/+}*), which has no discernible impact on PanINs. Expression levels of 42 validated Myc target genes (compiled from our own previous study in mouse⁵ and the human GSEA analysis of Myc gene targets from Dr Chi

Dang) are ranked from highest (top) to lowest in $KCR26^{MT2/MT2}$ pancreas and compared with expression of their “hypomorphic” $KCR26^{MT2/+}$ counterpart. It is immediately obvious that the responses of Myc target genes to a relatively modest 50% reduction in Myc level are very varied. Expression of most Myc target genes is decreased to some, variable, degree when Myc is activated at a reduced level. However, some Myc targets appear unaffected while some are no longer induced at all. Thus, sub-physiological levels of Myc elicit a qualitatively, as well as quantitatively, different pattern of transcriptional outputs. There are clearly many potential explanations for this varied Myc dependency across Myc target genes, such as variation in Myc binding affinity, differential Myc responsiveness of bound enhancers or promoters, or perhaps differential chaperoning by pioneer transcriptional co-factors but such analysis lies in the future. We also do not see any obvious biological rationale for which genes exhibit high versus low Myc-level dependence. To complement Supplementary Figure 5, we have compiled for the reviewers a more extensive heat map, including a fuller complement of 204 genes induced by physiological versus hypomorphic Myc levels in pancreatic epithelium and ranked based on their expression level in $KCR26^{MT2/MT2}$ Myc ON/Myc OFF ($\log_2FC > 0.5$ in $KCR26^{MT2/MT2}$ Myc ON/Myc OFF) (Rebuttal Figure 1). This further illustrates the notable features. We therefore now provide the entire transcriptomic data for the wider community (Accession number E-MTAB-10807).

The example of acutely imposed hypomorphism using reversibly switchable genetics that we here describe is, we submit, unprecedented and it has already yielded its fair share of surprising and unexpected observations. To go beyond this by undertaking an exhaustive analysis and search for other unknown deficits in the forcibly hypomorphed mice, as suggested by the reviewer, would constitute a huge open-ended fishing venture with no obvious point of resolution on its horizon, and is a long way away from identification of a Myc-sensitive early bottleneck in cancer evolution, which is the focus of our manuscript.

The reviewer also suggests changes in the sequence of primary and supplementary figures, to reinforce the impact of Myc hypomorphism on normal haematopoiesis. The problem here is that the many of the figures contain data that are then used at various points in the overall narrative of the manuscript. We are happy to discuss this with the Editor, should publication be agreed.

4. Figure 1B: histological analysis of lesions should include quantification of hyperplastic lesions vs. adenomas vs. adenocarcinomas. More details required in order to dissect effect on initiation vs. progression.

As is evident from Figure 1B, the problem here is that, aside from the rare p53-deficient lesions that sporadically “break” our hypomorphism mechanism, KRas^{G12D}-driven lesions generated in Myc hypomorphic mice are almost all indolent hyperplasias, with maybe a very few that might possibly be scored as grade 1. We have never seen any grade 2 or above lesions in hypomorphed animals so we can’t score them along the standard progression scale.

5. Figure 2A: histological analysis of lesions should include quantification of low- and high-grade PanINs. It is striking that the pancreas of hypomorphed mice seems to contain mucinous PanINs since the KPC model rarely develops these lesions (unlike the KP model),

suggesting an interaction with mutant p53. This merits some mechanistic analysis, especially considering the escapee tumors. Analysis of mutant p53 expression in the hypomorph mice would be interesting to understand the stalling of PanIN progression.

We understand the reviewer's point and have consulted our pathologist collaborator, Dr Mark Arends, regarding grading of PanINs. His view is that it is very hard to quantify high versus low grade PanINs as they are not really distinct lesions but on a continuous morphological spectrum and, moreover, there is no obvious underlying mechanistic basis for histological differences in PanIN grade. We therefore ask the reviewer to note that our analysis is focused on the requirement for wt Myc levels specifically at the transition from indolent PanIN to invasive neoplasm. We have no shred of evidence that Myc levels plays any role in histopathology and "grade" of pre-malignant PanIN progression.

As already outlined above, any potential role of p53 loss and/or potential gain of function by the *p53R172H* mutant in tumour evolution is obviously of great general interest. However, none of this bears on the subject of our manuscript, which is our discovered role played by Myc levels in early PanIN and LUAD progression. Our data clearly show that p53 status has no impact on the protective role of Myc hypomorphism *per se* (almost all the individual KRas^{G12D}-driven lesions in both lung and pancreas models, stall at the pre-malignant stage), but that p53 loss increases the chance of freak genetic accidents that wreck the *TRE-tTS^{KID}* mechanism our mouse model uses to induce Myc hypomorphism – the principal escape mechanisms being amplification of Myc expression and/or loss of expression of the tetracycline-regulated tTS^{KID} repressor (Supplementary Figure 4). While these occasional "broken" escapees rapidly expand and spread, the overwhelming majority of individual hypomorphed lesions remain stalled despite their p53 deficiency. Of course, once Myc hypomorphism mechanism has been compromised, this permits the transition from indolent to frank neoplasia and thenceforward, loss of p53 surveillance will greatly impact both trajectory and rapidity of subsequent tumour evolution. However, this is surely after the point in a manuscript focused on Myc-dependent events that we here and elsewhere^{6,7} have shown are independent of p53 mutation.

According to our pathologist collaborator Dr Mark Arends, the pancreata of hypomorphed mice do not exhibit mucinous PanINs; the pink proteinaceous secretion observed is protein and not mucin.

6. Figure 2 and others: The quantification using as only parameter the %Myc+ve cells is somewhat superficial.

The reviewer is concerned about our use of immunohistology to demonstrate elevated Myc levels. This may be "superficial" but, to be fair, we are dealing here with something of a side-show: i.e., the mechanism by which our reversibly inducible Myc hypomorphism mouse model gets broken by random mayhem in p53-defective lesions. In the cases shown, outgrowth of "escapee" tumours is almost always associated with either loss of expression of the tTS^{KID} repressor, or overt Myc over-expression – both of which nullify the imposed Myc hypomorphism. We previously reported that loss of p53 greatly increases genetic accidents that can lead to transgene loss or gene amplification⁸. Our purpose here is not to investigate how loss of p53 fosters genome and epigenome infidelity but to demonstrate that the only way that these pancreas pre-tumours ever progress is by wrecking the model's

mechanism for imposing Myc hypomorphism. To reflect the tangential nature of this “escapee” phenomenon (essentially, how our model can get broken) we have shifted the whole “escapee” description to a Supplemental Figure in the revised manuscript (Supplementary Fig. 4).

7. Figure 3. The fact that this lung cancer model is based on the Trp53 lox/lox alleles rather than in the mutant allele used in KPC mice renders the comparison with the pancreatic cancer model complex. The number of samples analyzed in panel 3B is very low.

We make no attempt to compare the lung and pancreas models. As already discussed above, the three cancer models we use all share KRas^{G12D} as the initial driver mutation but the manner and timing of KRas^{G12D} activation varies greatly. In the lung models Myc is hypomorphed before KRas^{G12D} activation – specifically, KRas^{G12D} is sporadically activated in lung epithelium of *adult*, pre-hypomorphed *adult* mice by Adeno-CRE inhalation. This generates multiple, independently evolving foci that sporadically transition to adenocarcinoma, presumably by aleatory acquisition of additional oncogenic mutations. The first (*MRK*) lung model is p53 is wild type, while in the second *MRK^{Pfl}* p53 is deleted concurrently with KRas^{G12D} activation. By contrast, in the PDAC *KPC* model, expression of KRas^{G12D}, together with the *R172H* p53 mutant, is driven *in embryos* from around E8 (*pdx/IPF1* promoter model) by Cre expression. Cre activation is not sporadic but driven throughout the embryonic pancreatic and duodenal progenitor cell population. In this pancreas model, Myc is hypomorphed long after initial KRas^{G12D} activation. Clearly, these models reflect a variety of different sequences of oncogene/tumour suppressor activation and status, as well as timing of Myc hypomorphic imposition. However, despite these differences, Myc hypomorphism completely blocks the transition from indolent pre-tumour to invasive adenocarcinoma in all these models (“escapee tumours are, as we show, no longer Myc hypomorphed).

As to the number of samples analyzed in the original Figure 3B (now supplementary Figure 4A), the number of mice might have been low but, as already noted, the number of individual lesions (focal hyperplasias in lung and PanINs in pancreas) is very high (now made clear by adding the number of individual lesions that ranged between 17 and 71 lesions in the figure legends of the revised Supplementary Figure 4). The number of individual lesions was too high to show on the graph. The original Figure 3B data were there solely to illustrate again the fact that the rare sporadic lesions that appear to escape Myc hypomorphism in actual fact result from a “broken” hypomorphism mechanism. The clear inference is that tumour progression in this p53-defective LUAD model remains potently blocked by Myc hypomorphism.

8. Figure 4. Validation of quantification of MycER expression in the current series of mice would be valuable, rather than relying on previous work.

We understand the reviewers point but the mouse models although in the experiments herein are identical to those used and published within just the past few years and, importantly, necessarily make use of data and inferences, such as comparative MycER levels, derived from those past experiments. A direct analysis of MycER levels in fibroblasts derived from the very same *R26^{MT2/MT2}* (2 copies *Rosa26*-driven MycER) and *R26^{MT2/+}* (1 copy

Rosa26-driven MycER) mice as those used in Figure 4 of the revised manuscript was shown previously in Figure 1A of ⁵. For the reviewer's benefit, we now provide Rebuttal Figure 3 providing evidence that insertion of the *TRE* into the endogenous Myc 2nd intron does not of itself measurably alter Myc expression levels:

- Rebuttal Fig. 3A: Scheme for serum induction of Myc in *wt* fibroblasts versus *Myc^{TRE}* fibroblasts.
- Rebuttal Fig. 3B: Identical kinetics of transient induction and levels of Myc induced in quiescent *wt* versus *Myc^{TRE}* fibroblasts by serum, showing not only that peak Myc levels are the same in each but so too is the classical autoregulation that Myc exhibits ⁹.
- Rebuttal Fig. 3C: Quantitation of Myc before (hypomorphed) versus after (*wt* level) addition of doxycycline, showing ~43% reduction of steady state Myc expression in log phase fibroblasts after hypomorphism is imposed.

In the revised manuscript (page 6) we now make specific mention of the fact that *TRE* insertion into the endogenous Myc 2nd intron has no measurable impact on Myc levels or kinetics of expression.

The "Tumor burden" parameter used for quantification is very crude; a finer assessment is important.

We agree that measurement of tumour burden is rather crude: on the other hand, the difference between tumour numbers in the *wt* versus the hypomorphs is pretty black and white – we *never* see any advanced tumours in hypomorphed animals (aside from the escapees that have broken Myc hypomorphism) in either our lung or pancreas cancer models.

It is proposed that CCL9 and IL-23 are required for the macrophage influx and lymphocyte depletion but neither of them is measured and there is no functional data allowing to conclude on the effects.

We are confused by this criticism. First, we don't just "propose" that CCL9 and IL-23 are required for, respectively the macrophage influx and lymphocyte depletion, in our lung adenocarcinoma model, we demonstrated it unambiguously ⁶. Moreover, in the previous incarnation of this manuscript, we directly showed (original Supplementary Figure 4B) that IL-23 is induced by "physiological" levels of MycER^{T2} but is no longer induced in *KRas^{G12D}*-driven pre-cancerous lung when Myc is expressed at sub-physiological "hypomorphic" levels. As it is clearly there in the original manuscript it is strange to be criticized for not showing it. We provide the IL-23 again in new version of the manuscript (Figure 4B, top right graph). We accept that CCL9 was not actually measured in the original manuscript – direct IHC measurement of CCL9 has continued to elude us using currently available antibodies. However, as outlined above, we previously demonstrated that CCL9 is the essential macrophage attractant through which Myc acts ⁶ so, in the present circumstances, macrophage influx seems to us to be a plausible surrogate for CCL9. In addition, we now also include new data (new Figure 4) that show that along with the failure of sub-physiological "hypomorphic" Myc levels to drive release IL-23 and drive influx of macrophages, we also see a failure to drive exclusion of T cells and NKp46⁺ NK-like cells

(new Figure 4A), both direct outcomes of IL-23 release⁶. Alongside, we present corresponding quantitation of individual lesions (new Figure 4B).

9. The transcriptome analysis shown in Supplementary Figure 5 is fragmentary and selective. A full analysis of the RNA-Seq data should be provided.

We have already discussed the new transcriptomics data we now present in the revised manuscript above (see new Supplementary Figure 5 and Reviewer's Figure 1).

Minor comments

1. The precise duration of "long term TMX diet" should be indicated since it is known that after long periods of administration can result in gastrointestinal damage.

The gastrointestinal damage resulting from extended Tamoxifen treatment at the dosage used occurs only after long periods. We don't really come close to that – we used 3 weeks of tamoxifen-containing diet in the pancreas and 6 weeks in the lung.

2. The unpaired t-test is not appropriate unless the data show a normal distribution, which requires a larger sample size. Therefore, Mann-Whitney should be used, instead.

Our resident statisticians (Cambridge Cancer Institute) recommend us to use Student's t test with Welch correction, which is more reliable when groups have unequal variances.

3. Review carefully for typos.

We have done our best!

Reviewer #2 (Remarks to the Author):

This is an interesting study and potentially important study that may help to instruct the development of anti-MYC therapy. However, some of these experiments are incremental advances over existing studies, and for some of the more original and creative parts of the manuscript, some interesting experiments remain to be performed or at least considered. In addition important papers in the literature need to be considered here. The authors need to stress more the experiments, conclusions and implications that are new versus those that are derivative.

Major points:

1. The system for hypomorphing MYC is interesting and potentially a very valuable tool. But there are other studies that need to be cited that show that in experimentally-engineered systems, MYC-haploinsufficiency decreases tumors (for example doi: 10.1101/cshperspect.a014290., Fig. 2) Yet, there are also uncited studies (doi:

10.1016/j.cell.2014.12.016.) that show that haploinsufficiency of MYC does not greatly reduce the incidence of spontaneously arising tumors, nor does this genetic haploinsufficiency it provoke any significant pathology during development and maturation. Is there not only a threshold for tumorigenesis, but a threshold for MYC-hypomorphism? If hypomorphic MYC is maintained at haploinsufficient levels, is it possible to protect from both tumors and bone marrow failure? The authors need to compare or at least discuss hypomorphic versus haploinsufficiency of MYC.

We guess this is in reference to two papers – ¹⁰ and ¹¹. It was, and always is, never our intention to fail to cite relevant literature, for which we unreservedly apologise. These papers consider cancer incidence in *Myc* haploinsufficient mice: the first considers APC-driven intestinal cancers and the second spontaneous lymphomas. The resistance of *Myc* haploinsufficient mice to intestinal neoplasia was shown initially in 2007 ¹² and in 2010 ¹³ and both of these sources were cited in the original manuscript. A delay in incidence of pancreatic adenocarcinoma in *Myc* haploinsufficient mice was reported in 2014 ¹⁴ and duly cited in the original manuscript. We are surely not guilty of failing to cite appropriate primary sources but are happy to add in these additional citations.

On a broader front, we go to great trouble to make the important point that germline haploinsufficiency cannot necessarily be presumed to phenocopy systemic *Myc* hypomorphism that is acutely imposed in adult mice. We also extensively discuss germline enhancer mutants of *Myc* that exhibit contextually reduced levels of *Myc* in certain circumstances and organs. The literature on adaptive compensation to germ-line genetic modification and its failure to safely predict the impact of adult imposition of the same - whether it be through switchable genes or pharmacological inhibition - is vast, if largely apocryphal, and is discussed extensively in the Discussion part of our manuscript (both original and revised). An apposite example of one such difference is that many of the embryonic lethal phenotypic attributes of total *Myc* deletion, including lethality, are consequences of *Myc* insufficiency in the placenta, not the embryo ¹⁵, a complication that is obviously irrelevant to adult manipulation of *Myc* expression. In addition, the known complexity of *Myc* transcriptional regulation, notably its autoregulation, raises the spectre that *Myc* dynamics may be very differently impacted when *Myc* expression is reduced by germ-line decrease in gene dosage versus direct, transient, inhibition of *Myc* activity in adult tissues. We are, after all, trying to model the consequences of direct, transient *Myc* inhibition in adult cancer prophylaxis.

2. Multiple papers—starting with Murphy, et al. (also doi: 10.1158/0008-5472.CAN-07-6192) have already suggested that a MYC-threshold must be exceeded for experimentally induced tumors. The MycERt experiments at little new to this study.

The reviewer argues that our MycER^{T2} experiments (original Figure 4, now Figures 3&4) add little new. However, there has been no previous analysis on defining exactly when, during the protracted evolution of cancers in mouse models, decreased levels of *Myc* impede tumour progression. This, alone, seems to us to be rather novel and important, and something that could never be predicted or gleaned from existing classical genetic studies.

The paper by Murphy *et al.* is well known to us as it came from our own laboratory. That paper analyzed biological outputs of Myc when expressed at different levels. However, this previous work says virtually nothing whatsoever about neoplasia, only about Myc's capacity to elicit ectopic proliferation and hyperplasia, and has recently followed up by our more extensive study of Myc's proliferative activity in different organs when expressed at different levels⁵. Also, the Murphy paper never investigated the impact of different levels of Myc expression on Myc's capacity to cooperate with other oncogenes, such as KRAs. The MycER^{T2} studies that we show demonstrate that *wt* physiological levels of Myc are both necessary (endogenous Myc) and sufficient (ectopic MycER^{T2}) to cooperate with KRAs to drive the transition from pre-tumour to invasive adenocarcinoma. They also show that reduced levels of both endogenous Myc and ectopic MycER^{T2} are both insufficient to support this transition.

The other 2008 paper mentioned by the reviewer,¹⁶ addresses a very different phenomenon – i.e. the levels of Myc required to maintain established Myc-driven lymphomas. Intriguing though this paper is, it tells us nothing directly about the role of Myc in early cancer evolution which, at risk of repetition, is the focus of this manuscript. To restate, our manuscript identifies a previously unknown Myc-dependent bottleneck, active at an extremely early stage of incipient tumour evolution, that has clear implications and relevance for early cancer detection and prevention, an inference accepted by both of the other reviewers.

3. The levels of MYC in the hypomorphic system are evaluated with and without doxycycline by immunostaining and qPCR, but are never compared to the levels of MYC in wild-type or haploinsufficient animals. With histologic sections, such comparisons always are subject to caveats concerning the relative contributions of different cell types in the selected samples and fields. It is important to compare the MR and wild-type mice to establish that the M allele is not itself somewhat hypomorphic. What is the output of the MR or M alleles compared with the native MYC allele? This is important because the existing literature indicates that MYC^{+/-} yields a relatively mild phenotype in most cases whereas MR mice exhibit more pronounced deficits in development and in hematopoiesis.

We have already discussed extensively the important differences between germline Myc haploinsufficiency and our adult imposed active Myc hypomorphism. Our switchable model clearly circumvents any adaptive compensation that may be driven by germ line Myc insufficiency and self-evidently bypasses the known impact of Myc haploinsufficiency on placenta¹⁵. The argument over which type of hypomorphism is “best” at most accurately depicting the impact of reduced Myc level on Myc biology could go on indefinitely and isn't the issue. The point of our manuscript is simply that adult imposition of Myc hypomorphism works as a cancer prophylactic by blocking an early transition in tumour evolution. No one is going to breed Myc haploinsufficient humans so validating the suppression of cancer by imposing reversible partial Myc suppression in adults is surely a far more realistic goal.

The reviewer's question about whether the mere insertion of a *TRE* in the Myc gene is sufficient to elicit Myc hypomorphism is entirely valid. We now present for the reviewers additional data (Rebuttal Figure 3) that directly compares not only the steady state levels of Myc in asynchronously proliferating Myc^{wt} versus Myc^{TRE/TRE} fibroblasts (Rebuttal Figure 3B)

but also shows normal kinetics of serum induction of the *wt* versus *TRE* Myc alleles (Rebuttal Figure 3A). As can be seen in Rebuttal Figure 3, *TRE* insertion alone has no measurable inhibitory impact on the transient peak of Myc in response to serum administration to quiescent cells, nor on the transient kinetics of that induction. A direct quantitative comparison of endogenous Myc levels in asynchronously growing mouse *Myc^{TRE/TRE}* fibroblasts indicates that activation of the tTS^{KID} repressor (by doxycycline withdrawal) decreases steady state Myc levels by around 43%.

4. In this light it would be good to evaluate whether MR/+ mice are tumor prone in the KRAS system. If there is a threshold for MYC in tumorigenesis, would MR/+ be above or below that threshold (MR/MR is clearly below it)? Also, use of homozygous MR mice protects from the usual genetic events that upregulate MYC in cancer. It would be interesting to observe in heterozygotes, how frequently and how efficiently genetic events up-regulate the wild-type MYC allele to bypass the MR system.

These are all worthy additional questions that might be investigated in the future but they do not speak to the critical early Myc-dependent threshold in cancer evolution that is the subject of this manuscript. The reviewer asks for literally years of additional work that might be “good to have” yet do not seem to us to be an essential underpinning of our manuscript. Moreover, it would need more than a “it would be good/interesting” to justify the extensive additional extra mouse studies.

5. It is hard to understand why the RosaMycERT/+ are “hypomorphic” with tamoxifen—don’t these mice also have two endogenous wild-type Myc alleles? Expression of RosaMycERT may be low relative to the endogenous allele, but how can the total be hypomorphic? Does this suggest that MYC doesn’t need to be hypomorphic, just not elevated to protect from tumorigenesis?

Here, there seems to be some confusion over gene copy number versus levels of endogenous Myc expression. As we painstakingly explain in the manuscript, adult pancreas and adult lung, like most adult organs, neither proliferate nor appreciably express endogenous Myc (see ⁵ and Supp. Figure 3). So yes, while there are indeed two endogenous *wt* Myc alleles present in these two tissues, neither is significantly expressed. Furthermore, Myc transcriptionally autoregulates to limit maximum levels and persistence of Myc in cells ⁹, with the consequence that excess ectopic Myc suppresses endogenous Myc expression.

6. It seems that escapees of the MR system lose the repressor. Are there other genetic events that bypass the hypomorphic system?

We also show that some escape by upregulating Myc (now supplementary Figure 4), However, our analysis of escape mechanism is far from exhaustive and there are literally hundreds of possible mechanisms by which our model might get broken. We can nonetheless account for nearly all escapee lung and pancreas tumours either through Myc upregulation and/or repressor loss, although how these actually arise is unclear. Myc can become upregulated in a wide variety of ways (e.g. insertional mutagenesis, enhancer mutation, gene or chromosome amplification, persistent upstream oncogenic signaling),

while loss of repressor expression might involve excision or methylation/silencing of the repressor transgene. Yet other, even more elaborate mechanisms are, of course, possible but seem to us to be rather tangential. One firm conclusion, however, is that our model only seems to break if p53 is absent from the outset. We never saw escapee lung tumours in the p53-competent KRas^{G12D} mouse model.

7. Does the inability of the MR- KRasG12D hyperplastic foci to invade or provoke a stromal response represent the lack of induction of a specific program by MYC or a general transcriptional downregulation lacking sufficient MYC?

We now provide a more detailed comparative analysis of transcriptional outputs of Myc at physiological versus hypomorphic levels (new Supplementary Figure 5) using the KRas^{G12D}-driven pancreas cancer model we previously described ⁷. Expression levels of 42 validated Myc target genes (compiled from our own previous study in mouse ⁵ and the human GSEA analysis of Myc gene targets from Chi Dang) are shown ranked from highest (top) to lowest in *KCR26*^{MT2/MT2} pancreas and compared with expression of their “hypomorphic” *KCR26*^{MT2/+} counterpart. The response of Myc target genes to a 50% reduction in Myc was very variable – some reduced to intermediate degrees, others unaffected, and yet other completely silent. Hence, sub-physiological levels of Myc elicit a qualitatively, as well as quantitatively, distinct pattern of transcriptional outputs. It is far too early to attribute any significance to this, except to say that Myc transcriptional output is clearly not necessarily linearly related to Myc expression level. To complement Supplementary Figure 5, we have compiled for the reviewers a heat map of 204 genes induced by physiological versus hypomorphic Myc levels in pancreatic epithelium (Rebuttal Figure 1). This further illustrates the notable features. We therefore now provide the entire transcriptomic data for the wider community (Accession number E-MTAB-10807).

Reviewer #3 (Remarks to the Author):

This paper presents a thorough and compelling analysis of the role that MYC expression levels play in tumor progression in mutant KRAS models of lung and pancreatic carcinoma. By cleverly deploying a tetracycline response element in a Myc intron, the authors are able to partially suppress endogenous Myc RNA expression by 20-50%, an effect readily reversible by addition of tetracycline. Their findings are striking in demonstrating that the decreased levels of Myc block the transition from hyperplasia to adenocarcinoma in both the lung and pancreas models. They go on to show that p53 is not required for the block and that "escaper tumors" tend to reestablish high levels of Myc, thus reinforcing the notion that full tumor growth is dependent on sustained higher Myc levels. Indeed, they show that metronomic up-down expression of Myc is sufficient to block tumorigenesis. Moreover, they provide experimental data implicating a failure of the Myc "hypomorphed" tumor to secrete cytokines likely to be involved in reconfiguring the stroma as required for progression.

Beyond the demonstration that Myc levels matter in tumor progression, the relevance of this paper is that it reinforces the concept of partial suppression of Myc expression as a potential therapeutic route. As the authors point out, the hypomorphed mice do not suffer from major

defects in development or growth. Therefore, if we could only figure out how to dial Myc down at an early stage in cancer progression we would have an effective, non-toxic, co-therapy.

Overall, I found the data presented to be very clear and convincing. My only quibble is a lack of quantitation relating to MYC protein levels. While several figures show IHC for MYC (e.g., Fig 2B) the difference in protein levels appeared rather marginal compared to the ISH. Perhaps the authors should consider including immunoblots from the relevant tissues.

We agree this whole issue of escaping tumours was indeed confusing as written in the original version of the manuscript. We apologise. The take home is quite straightforward:

1. Imposed Myc hypomorphism completely blocks progression (i.e. transition from pre-tumour to locally invasive cancer) KRas^{G12D}-driven cancers in lung. We never see any tumours outgrow the indolent AAH (Atypical adenomatous hyperplasia).
2. Broadly speaking, we also see the same profound block in KRas^{G12D}-driven pre-tumours in pancreas (PanIN) and lung (AAH) when p53 is functionally inactive (either deleted in lung, or mutated in pancreas). The great majority of lesions still remain stalled at the pre-tumour (PanIN or AAH) stage. Hence, p53 loss does not, of itself, circumvent the Myc hypomorphism block to tumour progression.
3. However, in the two p53-deficient models we do see occasional “escapee” tumours emerge from the background sea of stalled lesions.
4. Examination of these rare and sporadic “escapee” tumours reveals that almost all of them have broken the hypomorphism mechanism used by our model: specifically, they either silence expression of the tTS^{Kid} repressor that induces Myc hypomorphism or they upregulate Myc so that Myc expression is no longer hypomorphic. We never see either of these two events occurring in the p53-proficient version of the lung model.
5. We have seen analogous GEMM fragility in other p53-deficient mouse models we have generated: it appears that loss of p53 fosters orthotopic gene and transgene instability, which is no surprise.
6. In the original manuscript, we erred in spending a lot of time discussing this phenomenon (the original Figure 2) when it is actually tangential to our discovery of the early pre-tumour-to-cancer transition bottleneck over which Myc presides. However, we did this to underscore that p53-deficient escapee tumours do not escape *in spite* of Myc’s being hypomorphed, they escape *because Myc hypomorphism is no longer working*. Hence, the Myc hypomorphic block *is not* p53-dependent.

In the revised manuscript, we have now appropriately relegated the whole “escapee” sub-narrative to Supplementary Figure 4 and simply provide evidence of the two escape mechanisms in representative examples of “escaping” lung (Supplementary Fig 4A) and pancreas (Supplementary Fig 4B) tumours. The details of these studies and their implications are comprehensively described in the relevant text (pages 8-9 and 15-16).

REFERENCES

- 1 Westcott, P. M. *et al.* The mutational landscapes of genetic and chemical models of Kras-driven lung cancer. *Nature* **517**, 489-492, doi:10.1038/nature13898 (2015).
- 2 Muncan, V. *et al.* Rapid Loss of Intestinal Crypts upon Conditional Deletion of the Wnt/Tcf-4 Target Gene c-Myc. *Molecular and Cellular Biology* **26**, 8418-8426, doi:papers3://publication/doi/10.1128/MCB.00821-06 (2006).
- 3 Delgado, M. D. & Leon, J. Myc roles in hematopoiesis and leukemia. *Genes & Cancer* **1**, 605-616, doi:papers3://publication/doi/10.1177/1947601910377495 (2010).
- 4 Wilson, A. *et al.* c-Myc controls the balance between hematopoietic stem cell self-renewal and differentiation. *Genes Dev* **18**, 2747-2763 (2004).
- 5 Bywater, M. J. *et al.* Reactivation of Myc transcription in the mouse heart unlocks its proliferative capacity. *Nat Commun* **11**, 1827, doi:10.1038/s41467-020-15552-x (2020).
- 6 Kortlever, R. M. *et al.* Myc Cooperates with Ras by Programming Inflammation and Immune Suppression. *Cell* **171**, 1301-1315.e1314, doi:papers3://publication/doi/10.1016/j.cell.2017.11.013 (2017).
- 7 Sodir, N. M. *et al.* MYC Instructs and Maintains Pancreatic Adenocarcinoma Phenotype. *Cancer Discovery* **10**, 588-607, doi:papers3://publication/doi/10.1158/2159-8290.CD-19-0435 (2020).
- 8 Soucek, L. *et al.* Inhibition of Myc family proteins eradicates KRas-driven lung cancer in mice. *Genes & Development* **27**, 504-513, doi:papers2://publication/doi/10.1101/gad.205542.112 (2013).
- 9 Penn, L. J., Brooks, M. W., Laufer, E. M. & Land, H. Negative autoregulation of c-myc transcription. *EMBO J* **9**, 1113-1121, doi:papers3://publication/doi/10.1093/emboj/cda066 (1990).
- 10 Wiese, K. E. *et al.* The role of MIZ-1 in MYC-dependent tumorigenesis. *Cold Spring Harb Perspect Med* **3**, a014290, doi:10.1101/cshperspect.a014290 (2013).
- 11 Hofmann, J. W. *et al.* Reduced Expression of MYC Increases Longevity and Enhances Healthspan. *Cell* **160**, 477-488, doi:papers3://publication/doi/10.1016/j.cell.2014.12.016 (2015).
- 12 Yekkala, K. & Baudino, T. A. Inhibition of intestinal polyposis with reduced angiogenesis in ApcMin/+ mice due to decreases in c-Myc expression. *Mol. Cancer Res.* **5**, 1296-1303, doi:papers3://publication/doi/10.1158/1541-7786.MCR-07-0232 (2007).
- 13 Athineos, D. & Sansom, O. J. Myc heterozygosity attenuates the phenotypes of APC deficiency in the small intestine. *Oncogene* **29**, 2585-2590, doi:papers3://publication/doi/10.1038/onc.2010.5 (2010).
- 14 Walz, S. *et al.* Activation and repression by oncogenic MYC shape tumour-specific gene expression profiles. *Nature*, 1-18, doi:papers3://publication/doi/10.1038/nature13473 (2014).
- 15 Dubois, N. C. *et al.* Placental rescue reveals a sole requirement for c-Myc in embryonic erythroblast survival and hematopoietic stem cell function. *Development* **135**, 2455-2465, doi:10.1242/dev.022707 (2008).
- 16 Shachaf, C. M. *et al.* Genomic and proteomic analysis reveals a threshold level of MYC required for tumor maintenance. *Cancer Research* **68**, 5132-5142, doi:papers3://publication/doi/10.1158/0008-5472.CAN-07-6192 (2008).

Rebuttal Figures for Reviewers

Rebuttal Figure 1: Deregulated sub-physiological levels of Myc retain measurable but selective transcriptional activity

Heat map of RNA expression (RNA-seq) in pancreas samples harvested from 12 week-old *KCR26M^{MT2/MT2}* and *KCR26M^{MT2/+}* mice where Myc was either activated for 12 hours (Myc ON) or never activated (Myc OFF). The column on the right shows induction of 204 genes by two copies of *Rosa26-MycER^{T2}* in *KCR26M^{MT2/MT2}* pancreas (equivalent to physiological Myc levels) relative to no Myc activation control (Myc ON/Myc OFF). This is compared (left) with Myc ON/Myc OFF induced in *KCR26M^{MT2/+}* mice (one copy of *Rosa26-MycER^{T2}* equivalent to hypomorphic levels of Myc). Genes are ranked based on their expression level in *KCR26M^{MT2/MT2}* Myc ON/Myc OFF ($\log_2FC > 0.5$ in *KCR26M^{MT2/MT2}* Myc ON/Myc OFF). n=4 mice per group.

**Sodir et al. Rebuttal Figure 1:
Deregulated low levels of Myc retain measurable but selective transcriptional activity**

Rebuttal Figure 2: Hypomorph-equivalent sub-physiological levels of deregulated Myc lie below the threshold required to engage instructive signals necessary to drive progression from indolent pre-tumour to invasive neoplasia

Quantification of CD206, IL23, CD3 and NKp46 immunohistochemical staining (IHC) in sections of lungs harvested from *KR26^{MT2/MT2}* and *KR26^{MT2/+}* mice 12 weeks after activation of *KRas^{G12D}* either without or with activation of *MycER^{T2}* for 3 days. Results depict mean \pm SD of independent tumours (small symbols) with 3-6 mice per treatment group (large symbols). IL23+ staining intensity was normalized to the number of nuclei per FOV (Field of view) as described in ¹. For NKp46 staining, only tumours connected to clearly distinguishable vascular and airway regions were considered; the numbers of tumour-associated NKp46+ cells per tumour per lung section were counted. Data were analyzed using unpaired t- test with Welch's correction (CD06, IL23, CD3) or two-ANOVA (NKp46). **p < 0.01, ***p < 0.001, ****p < 0.0001, ns= non-significant. SD = standard deviation.

- 1 Crowe, A. R. & Yue, W. Semi-quantitative Determination of Protein Expression using Immunohistochemistry Staining and Analysis: An Integrated Protocol. *Bio Protoc* **9**, doi:10.21769/BioProtoc.3465 (2019).

Sodir *et al.* Rebuttal Figure 2:

Hypomorph-equivalent sub-physiological levels of deregulated Myc lie below the threshold required to engage instructive signals necessary to drive progression from indolent pre-tumour to invasive neoplasia

Rebuttal Figure 3: *TRE* insertion into the endogenous *c-myc* gene 2nd intron does not measurably impact endogenous Myc expression level or kinetics of serum regulation

A: Scheme for analysis of mitogenic induction of Myc. Mouse adult lung fibroblasts (MALFs) were isolated from *Myc^{TRE/TRE}* or *wt* adult mice and maintained in normal medium (DMEM with 10% FBS). Prior to mitogen induction, MALFS were cultured in 0.1% FBS for 72 hrs to induce quiescence and then serum-stimulated (10% FBS) for 0, 2, 6 and 12 hrs.

B: Kinetics of serum induction of Myc in quiescent control (*wt*) versus *Myc^{TRE/TRE}* MALFS, showing identical signature transient Myc induction. Cells were collected and lysed in RIPA buffer and whole protein lysates (5 μ g) were subjected to Western blot analysis for Myc ab32072.

C: Left: Comparison of Myc levels in asynchronous log-phase *Myc^{TRE/TRE};tTS^{Kid/-}* MALFs growing in either the absence (hypomorphed) or presence (non-hypomorphed) of 1mg/ml of doxycycline

Right: ImageJ quantification of relative Myc protein levels (normalized to β -actin loading control) in asynchronous growing *Myc^{TRE/TRE};tTS^{Kid/-}* MALFS in the absence or presence of doxycycline.

Sodir *et al.* Rebuttal Figure 3:

TRE insertion into the endogenous *c-myc* gene 2nd intron does not measurably impact endogenous Myc expression level or kinetics of serum regulation

A

B

C

REVIEWERS' COMMENTS

Reviewer #1 (Remarks to the Author):

The authors have satisfactorily responded to my comments and the manuscript is ready for publication.

Reviewer #2 (Remarks to the Author):

The revised manuscript by Sodir and colleagues is improved and has satisfactorily addressed *most* of my concerns as well as, I believe, the issues raised by the other referees.

There is, however, one issue persists as raised in my previous **point 5**-notwithstanding the authors' somewhat quarrelsome rebuttal. Basically, the issue is what level of Myc expression defines hypomorphism and prevents tumorigenesis and tumor progression. There is no question that the innovative strategies of these investigators provide a proof-of-principle that restricting Myc levels is protective against several types of cancer. Their RosaMycER^{T2}/+ results suggest that preventing Myc over-expression (is this hypomorphism?) is sufficient for anti-cancer prophylaxis as opposed to enforced, hypomorphic Myc-under-expression. Despite their attestations, Myc expression is detectable in organs such as lung and pancreas; many databases document this. (This baseline expression may be unimportant in the absence of additional stress, but perhaps helps to keep the gene on standby for future challenges). Similarly, the argument about Myc autoregulation is specious; Myc autorepression only occurs at high, oncogenic levels of Myc (first found in Burkitt lymphoma). In single cells, under physiological, non-neoplastic conditions, Myc expression is bi-allelic. Adding a single MycER^{T2} *on top* of baseline-Myc in mature tissues provides an insufficient impulse for tumorigenesis and argues for an oncogenic threshold. But the authors clearly establish that maneuvers which limit upward excursions of Myc or restrict its activity, even incrementally, may prove profoundly salutary.

The authors should mention that the decrement of Myc required for hypomorphic protection has not been established.

This is an important and well-executed study.

Reviewer #3 (Remarks to the Author):

I've read through the rebuttal letter and the revised manuscript and feel that the authors have adequately addressed the major comments of all three reviewers. I therefore recommend publication of the revised paper.

I suggest that authors include rebuttal fig 3 C in supplementary data.